# In vivo spatiotemporal control of voltage-gated ion channels by using photoactivatable peptidic toxins

Jérôme Montnach [1,2], Laila Ananda Blömer[2,3], Ludivine Lopez[1,2,4], Luiza Filipis [2,3], Hervé Meudal [5], Aude Lafoux[6], Sébastien Nicolas[1,2], Duong Chu[7], Cécile Caumes[4], Rémy Béroud[4], Chris Jopling[8], Frank Bosmans[9], Corinne Huchet[6], Céline Landon [5], Marco Canepari[2,3] & Michel De Waard [1,2,4 ✉]

Photoactivatable drugs targeting ligand-gated ion channels open up new opportunities for light-guided therapeutic interventions. Photoactivable toxins targeting ion channels have the potential to control excitable cell activities with low invasiveness and high spatiotemporal precision. As proof-of-concept, we develop HwTxIV-Nvoc, a UV light-cleavable and photo-activatable peptide that targets voltage-gated sodium (Na$_V$) channels and validate its activity in vitro in HEK293 cells, ex vivo in brain slices and in vivo on mice neuromuscular junctions. We find that HwTxIV-Nvoc enables precise spatiotemporal control of neuronal Na$_V$ channel function under all conditions tested. By creating multiple photoactivatable toxins, we demonstrate the broad applicability of this toxin-photoactivation technology.

[1] l'institut du thorax, INSERM, CNRS, UNIV NANTES, F-44007 Nantes, France. [2] Laboratory of Excellence Ion Channels, Science & Therapeutics, F-06560 Valbonne, France. [3] Laboratoire Interdisciplinaire de Physique, Université Grenoble Alpes, CNRS UMR 5588, 38402 St Martin d'Hères, cedex, France. [4] Smartox Biotechnology, 6 rue des Platanes, F-38120 Saint-Egrève, France. [5] Center for Molecular Biophysics, CNRS, rue Charles Sadron, CS 80054, Orléans 45071, France. [6] Therassay Platform, IRS2-Université de Nantes, Nantes, France. [7] Queen's University Faculty of Medicine, Kingston, ON, Canada. [8] Institut de Génomique Fonctionnelle, 141 rue de la Cardonille, 34094 Montpellier, France. [9] Department of Basic and Applied Medical Sciences, Ghent University, Ghent, Belgium. ✉email: michel.dewaard@univ-nantes.fr

Ion channels are pore-forming transmembrane proteins that allow the regulated flow of cations or anions across membranes. Due to their important biological roles in many cell types, ion channels constitute drug targets for the treatment of diseases, such as type-2 diabetes, hypertension, epilepsy, chronic pain, cardiac arrhythmia, and anxiety, and many are part of the classical drugs on the WHO's list of essential medicines (https://list.essentialmeds.org). Ion channels are regarded as difficult targets for drugs due to the challenge to achieve subtype selectivity and because they represent complex protein structures embedded in the plasma membrane. Biological compounds, such as peptides found in animal venoms, demonstrate their usefulness in reaching high selectivity and affinity towards their targets owing to chemical surfaces larger than those of small organic compounds. As such, they represent exquisite high affinity and selective tools for the pharmacological control of ion channels and of cellular excitability[1] and play a vital role in the deorphanization and classification of ion channels[2]. While animal peptide toxins originate from a wide variety of venomous species, they have in common a compact 3D structure, largely imposed by internal disulphide bridges that promote the formation, organization, and stabilization of secondary structures[3]. Mechanistically, peptide toxins targeting ion channels can inhibit the pore or modify the gating process to alter channel activation or inactivation properties and thereby act as inhibitors or activators[4]. Another interesting feature is the diversity in selectivity encountered so far. Some venom peptides such as ω-conotoxin-GVIA (N-type $Ca_V2.2$ channel[5]), BeKm1 (hERG channel[6]) or μ-conotoxin KIIIA ($Na_V1.2$[7]) are selective for a particular target. Others bind to virtually an entire class of ion channels (i.e. AaHII and $Na_V$ channels[8]). Hence, venom peptides appear as a promising class of compounds to control the activity of excitable cells with a user-defined selectivity.

When it comes to provide a light-mediated control of cell excitability, the technology developed the furthest is optogenetics as witnessed by the substantial amount of literature on the topic[9,10]. Undoubtedly, optogenetics have transformed many areas of biological research demonstrating the usefulness for a precise control of biological functions thanks to the unsurpassed spatial and temporal resolution of focussed light. However, this technology does suffer from several drawbacks that are challenging to overcome for therapeutic applications, except on rare occasions, such as recently shown for vision treatment[11]. Optogenetics is largely based on the use of genetically encoded material that needs to be supplied externally for invasive modification of the cell proteome. In contrast, photopharmacology does not require any modification of the cell proteome or genetic background and responds more favorably to the exigences of regulatory agencies for therapeutic development. Likewise to optogenetics, photoactivatable caged compounds are powerful tools for modulating the function of native proteins with high spatiotemporal resolution. This concept has shown promise in animal models for vision restoration[11] and pain management[12]. Although uncaging of chemicals for ligand-gated channels enabled seminal optopharmacology studies informing on their function and subcellular location in complex biological environments[13–16], caged ligands targeting voltage-gated ion channels more specifically remain rare. The example of saxitoxin demonstrates, however, that it is an interesting axis of development[17]. It is this challenge to deliver a greater number of interesting ligands amenable to light control that our project addresses. Indeed, with recent progresses in chemistry, it is now feasible to transform a venom peptide into a photosensitive tool. The most straightforward chemical strategy to achieve this aim is to add a photosensitive cage on the side chain of a residue that is key for activity[18–21]. Actually, a shared feature of most peptide toxins is that activity generally relies on a limited set of amino acids. Typically, a Lys residue (or a residue that can be substituted by Lys), is often critical for biological activity[22]. A similar approach has been used in the past on Cys residues to control region-selective disulfide bonding of oxytocin[23], α-conotoxin IMI[23], and human insulin[23], but not to our knowledge to endow the pharmacological action of structurally complex toxins with a photosensitivity property. Based on our comprehension of peptide pharmacophores, we present herein a general caging strategy, involving covalent attachment of an orthogonal o-nitroveratryloxycarbonyl (Nvoc) protecting group on the lateral chain of a Lys residue that remains compatible with solid-phase peptide synthesis and that can be cleaved by photolysis under physiological conditions. We use both brain slices and in vivo experiments (mice neuromuscular junction and zebrafish larvaes) to validate the fact that illumination leads to selective inhibition of voltage-gated $Na^+$ ($Na_V$) channel with high spatial resolution. Finally, we illustrate that the technique can be generalized to peptide toxins possessing more or less ion channel selectivity, and is applicable to both inhibitors and activators.

## Results

**Design, chemical synthesis, and structure of HwTxIV-Nvoc.** To demonstrate the applicability of modifying a venom peptide into a photosensitive tool in spite of the structural complexity and presence of multiple disulfide bridges, we synthetized a pure HwTxIV-Nvoc compound based on a highly effective analogue of native HwTxIV (nHwTxIV) (Fig. 1a). nHwTxIV was first identified as a $Na_V1.7$ inhibitor but shown recently to target also $Na_V1.6$ with a lower affinity[24]. We first aimed to develop a HwTxIV analogue with convergent blocking potencies for $hNa_V1.1$, $hNa_V1.2$, $hNa_V1.6$, and $hNa_V1.7$ channels. Based on earlier structure-function analyses using nHwTxIV single amino acid mutations[25–27] and several 3D structures of HwTxIV-$Na_V1.7$ complex (Supplementary Fig. 1a)[28–30], several residues were targeted to produce a more general and potent $Na_V$ channel inhibitor. By synthesizing 20 analogues and evaluating their potencies by building dose-response curves thanks to an automated patch-clamp setup on $hNa_V1.1$, $hNa_V1.2$, $hNa_V1.6$ and $hNa_V1.7$ channels, we determined that the nHwTxIV analogue ($HwTxIVG^1G^4K^{36}$ that we shall term HwTxIV afterwards for simplicity) was the most potent for a pan-$Na_V$ action (equivalent potencies on several $Na_V$ channels evaluated as $IC_{50}$ values for inhibition of peak $Na^+$ currents in the whole-cell configuration; $hNa_V1.1$: 18.9 ± 1.2 nM, $n = 3$-10 cells per data point vs. 50.7 ± 1.2 nM for nHwTxIV, $n = 3$–6; $hNa_V1.2$: 11.9 ± 1.1 nM, $n = 4$–9 vs. 8.3 ± 1.2 nM for nHwTxIV, $n = 6$–11; $hNa_V1.6$: 19.5 ± 1.3 nM, $n = 6$–10 vs. 154.3 ± 1.2 nM, $n = 3$–7 for nHwTxIV; $hNa_V1.7$: 15.2 ± 1.1 nM, $n = 7$–17 vs. 9.9 ± 0.9 nM, for nHwTxIV, Supplementary Fig. 1b and Supplementary Table 1). The improvement was most remarkable for $hNa_V1.6$, opening applications for this peptide. By mutating only the $K^{32}$ into $N^{32}$ ($IC_{50} > 1$ μM, $n = 4$–7 for nHwTxIV-$N^{32}$ on $hNa_V1.7$ vs. $IC_{50} = 9.9$ nM ± 0.9 nM, $n = 5$–6 for nHwTxIV on $hNa_V1.7$), we confirmed that the $K^{32}$ residue is an ideal amino acid residue for chemical modifications aimed at reducing HwTxIV potency[30,31] (Supplementary Fig. 1c). Since $N^{32}$ is sterically similar to $K^{32}$, these results indicate the importance of the positive charge carried by $K^{32}$ for activity. Before proceeding with the chemical synthesis of HwTxIV-Nvoc, we noted that $K^{32}$ interacts with $E^{811}$, $D^{816}$, and $E^{818}$ of the loop between S3 and S4 of $Na_V1.7$-NavAb (pdb code 7K48)[30]. Based on those observations, we confirmed that, if the binding of HwTxIV-Nvoc onto the channel had to occur similarly to the non-caged HwTxIV, the presence of the Nvoc protecting group would introduce steric clashes that predictably should

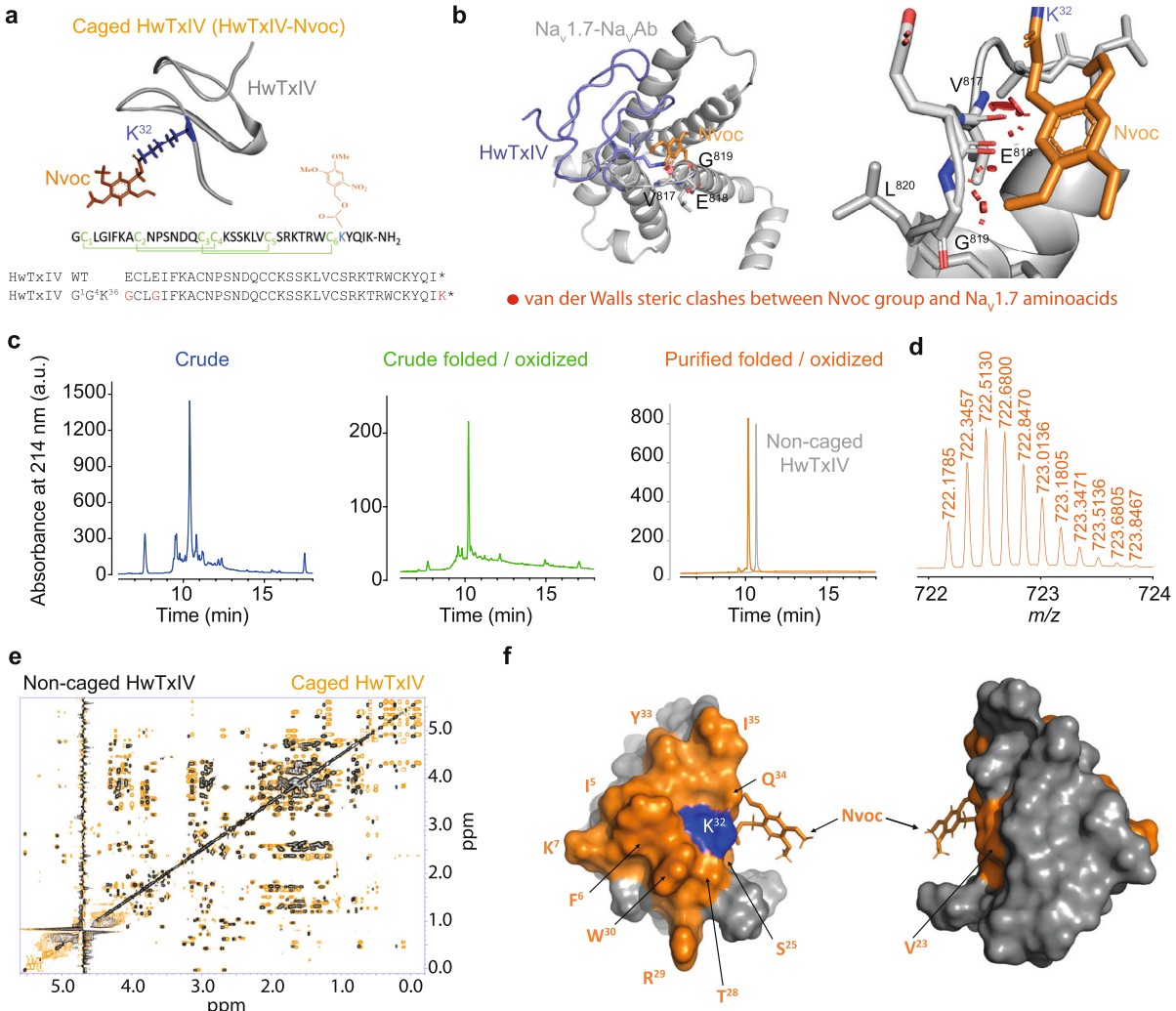

**Fig. 1 Engineering of photoactivatable HwTxIV-Nvoc to modulate Na$_V$ channels. a** Structure of HwTxIV-Nvoc with K$^{32}$ and Nvoc group reported in blue and orange sticks respectively. HwTxIV structure has been isolated from PDB 7K48. The sequence chosen for caging is shown upon sequence alignment with nHwTxIV. **b** Illustration of steric van der Walls clashes (red cylinders) resulting from the addition of the Nvoc protecting group onto K$^{32}$ upon binding of HwTxIV onto the chimeric Na$_V$1.7-Na$_V$Ab in the position as published[30] (PDB code 7K48). **c** RP-HPLC elution profiles of crude, crude folded/oxidized, and purified folded/oxidized HwTxIV-Nvoc. RP-HPLC elution profile of purified folded/oxidized non-caged HwTxIV has been superimposed for comparison. **d** LC-ESI QTOF analysis of HwTxIV-Nvoc denotes an exact mass of 722.1785 [M + 6H]$^{6+}$. **e** Superimposition of TOCSY spectra (sidechains area only for clarity) of the non-caged HwTxIV (in black), and of HwTxIV-Nvoc before illumination (in orange). **f** Residues showing major chemical shifts modifications in the presence of Nvoc are reported in orange on the surface of the molecule (180°views). K$^{32}$ is highlighted in blue. The Nvoc group is reported in sticks. Source data are provided as a Source Data File.

reduce peptide affinity (Fig. 1b). Similarly, the masking of the protonated lysine as an electrostatically neutral carbamate, also removes the positive charge that is suspected to contribute to toxin activity. HwTxIV-Nvoc was assembled stepwise using Fmoc-based SPPS on a 2-chlorotrityl-polystyrene resin, preloaded with a rink amide linker. The crude peptide was obtained in quantitative yield and of good quality to be directly oxidized. The presence of the Nvoc protecting group on K$^{32}$ which is located next to the sixth Cys residue within the HwTxIV sequence can compromise the proper assembly of the three disulfide bridges within an ICK fold. The folding process was followed by analytical RP-HPLC and showed completion after 16 h with the disappearance of the unfolded peptide and formation of one main oxidation peak occurring with a slight left shift on the elution profile. Finally, the pH of the reaction mixture was adjusted to three and the oxidized peptide was purified to homogeneity by preparative RP-HPLC. Two successive purifications, first using a C18 (5 μm, 130 Å) Waters CSH column, second using a C18

(10 μm, 100 Å) Phenomenex Luna stationary phase (eluent system H$_2$O/MeCN + 0.1% TFA), were needed to complete production of pure HwTxIV-Nvoc with a reasonably good global yield of 8.2% including SPPS, folding and purifications (Fig. 1c). The peptide was obtained with a measured exact mass of 722.1785 [M + 6H]$^{6+}$ by LC-ESI QTOF (calculated mass of 4327.08 Da, which is consistent with the theoretical monoisotopic mass of 4327.00 Da) (Fig. 1d). The proper mass indicates that the Nvoc protecting group was not removed by the TFA cleavage treatment of the synthetic crude unfolded peptide or during oxidative folding. As attested by the dispersion of the resonances, noncaged HwTxIV is well structured (Fig. 1e black spectra, and Supplementary Data 1). As expected, the presence of the aromatic Nvoc group on K$^{32}$ for HwTxIV-Nvoc disturbs the chemical environments of a series of residues on the side of the molecule where K$^{32}$ is located without affecting the global fold of the peptide (Fig. 1e, f, orange spectra). The presence of the grafted Nvoc group generates a doubling of certain peaks, possibly

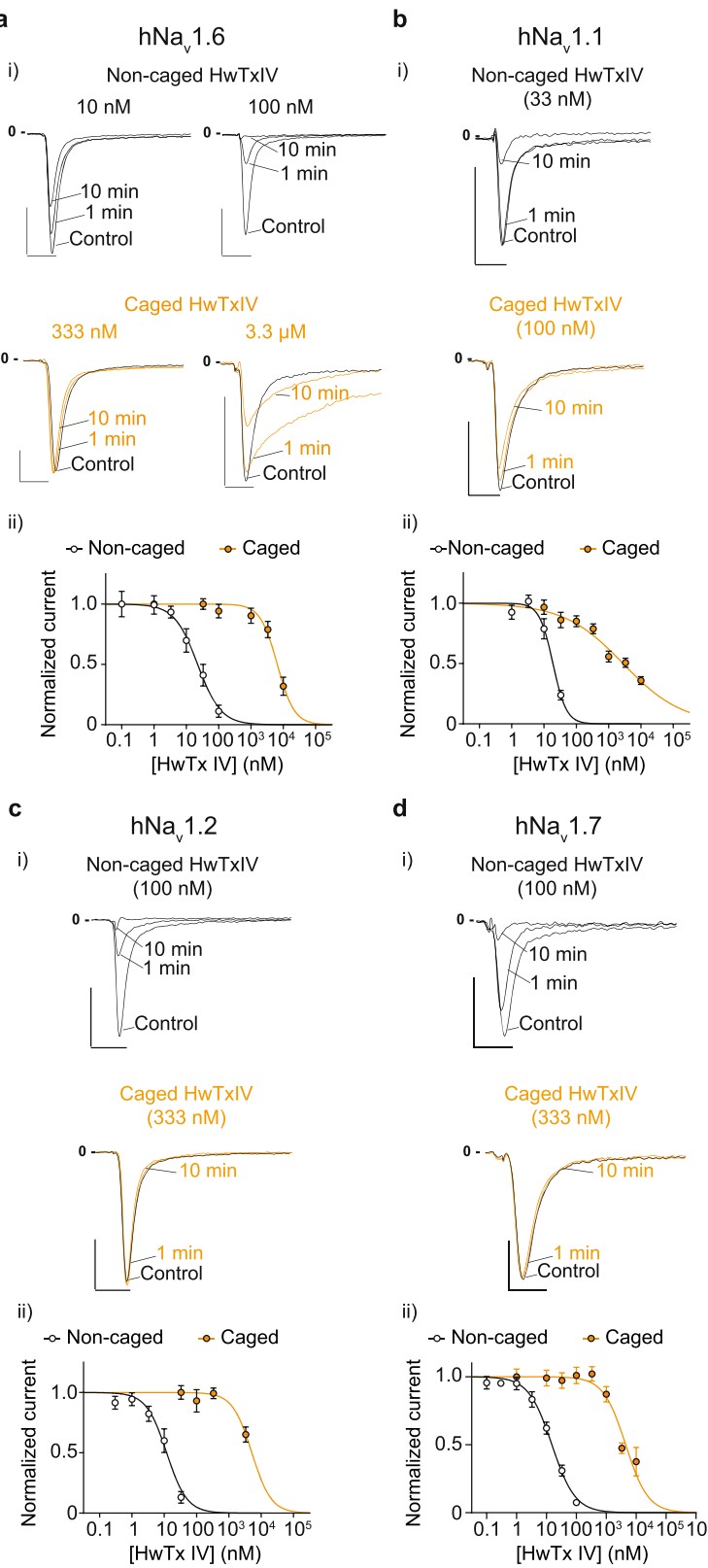

indicating different orientations of the Nvoc protecting group. HwTxIV-Nvoc showed excellent dark stability in solution for over 1 week (Supplementary Fig. 2).

**HwTxIV-Nvoc drastically reduces the potency of the peptide for Na_V channels**. Next we validated functionally that the steric

clashes introduced by grafting the Nvoc group on HwTxIV combined with the elimination of $K^{32}$ positive charge is sufficient to reduce the Na_V channel blocking potency of the peptide. Compared to non-caged HwTxIV, the caged toxin induces a 292- and 293-fold increase of the $IC_{50}$ value on hNa_V1.6 and hNa_V1.7, respectively (Fig. 2a, d) and even greater shifts are observed on hNa_V1.1 and hNa_V1.2 (Fig. 2b, c). Concentrations above 1 μM are

**Fig. 2 Caging of HwTxIV drastically reduces affinity of HwTxIV for hNa$_V$ channels. a** i) Representative recording of hNa$_V$1.6 currents elicited at 10 mV from a holding potential of -100 mV illustrating the extent of current block by different concentrations of noncaged HwTxIV (black) and HwTxIV-Nvoc (orange). ii) Average dose-response curves for hNa$_V$1.6 current inhibition by non-caged HwTxIV (black) and HwTxIV-Nvoc (orange). The data were fitted according to a Hill equation (non-caged HwTxIV IC$_{50}$ = 22.4 nM (n = 5) and HwTxIV-Nvoc IC$_{50}$ = 6539 nM (n = 7) (292-fold reduction in IC$_{50}$)). Scales: 2 ms, 1 nA. Note that on hNa$_V$1.6, high concentrations of HwTxIV-Nvoc produce a slowing of channel inactivation due to binding on a low-affinity site absent on hNa$_V$1.2 (Supplementary Figure 3). **b** i) Representative recording of hNa$_V$1.1 current elicited at 10 mV from a holding potential of -100 mV illustrating the extent of current block by different concentrations of non-caged HwTxIV (black) and HwTxIV-Nvoc (orange). ii) Average dose-response curves for hNa$_V$1.1 current inhibition by noncaged HwTxIV (black) and HwTxIV-Nvoc (orange). The data were fitted according to a Hill equation (noncaged HwTxIV IC$_{50}$ = 3.60 nM (n = 12); HwTxIV-Nvoc IC$_{50}$ = 2794 nM (n = 8) (776-fold reduction in IC$_{50}$)). Scales: 2 ms, 1 nA. **c** i) Representative recording of hNa$_V$1.2 current elicited at 10 mV from a holding potential of -100 mV illustrating the extent of current block by different concentrations of noncaged HwTxIV (black) and HwTxIV-Nvoc (orange). Scales: 2 ms, 1 nA. ii) Average dose-response curves for hNa$_V$1.2 current inhibitions by non-caged HwTxIV (black) and HwTxIV-Nvoc (orange). The data were fitted according to a Hill equation (noncaged HwTxIV IC$_{50}$ = 3.23 nM (n = 11); HwTxIV-Nvoc IC$_{50}$ = 5141 nM (n = 6) (1592-fold reduction in IC$_{50}$)). **d** i) Representative recording of hNa$_V$1.7 current elicited at 10 mV from a holding potential of -100 mV illustrating the extent of current block by different concentrations of non-caged HwTxIV (black) and HwTxIV-Nvoc (orange). Scales: 2 ms, 0.5 nA. ii) Average dose-response curves for hNa$_V$1.7 current inhibitions by non-caged HwTxIV (black) and HwTxIV-Nvoc (orange). The data were fitted according to a Hill equation (non-caged HwTxIV IC$_{50}$ = 15.23 nM (n = 92 cells); HwTxIV-Nvoc IC$_{50}$ = 4467 nM (n = 83 cells) (293-fold reduction in IC$_{50}$)). All data are presented as mean ± SEM. Source data are provided as a Source Data File.

needed for HwTxIV-Nvoc to start exhibiting inhibition of all Na$_V$ channel subtypes tested. Remarkably, the toxin also slows the fast inactivation of hNa$_V$1.6 (Fig. 2a, Supplementary Fig. 3a,b). We interpret this effect as due to the existence of a low-affinity binding locus on domain IV voltage sensor[4] that is now revealed by the absence of a high-affinity block due to Nvoc presence. To test this hypothesis, we transplanted the S3-S4 motif from each of the four voltage-sensor domains (VSDI-IV) of hNa$_V$1.6 into the homotetrameric rK$_V$2.1 channel according to previously described boundaries[32–34]. The transferred region in all of the functional chimeras contains the crucial basic residues that contribute to gating charge movement in K$_V$ channels[35–37]. Examination of conductance-voltage (G-V) relationships for the hNa$_V$1.6/rK$_V$2.1 chimeras revealed that each of the four voltage-sensor motifs has a distinct effect on the gating properties of K$_V$2.1. We next examined the effect of HwTxIV-Nvoc on the Na$_V$1.6/K$_V$2.1 VSD chimeras. 5 µM toxin did not alter the G-V relationship of the VSDI and VSDIII chimera, whereas VSDII and VSDIV were clearly inhibited (Supplementary Fig. 3c). HwTxIV-Nvoc binding to VSDII typically results in Na$_V$ channel inhibition, whereas VSDIV is involved in channel fast inactivation[32,37] although a role in channel opening may also be possible[38,39]. Thus, we conclude that, above 1 µM, HwTxIV-Nvoc influences hNa$_V$1.6 gating primarily by interacting with the S3-S4 motif in VSDII and VSDIV.

**UV-dependent uncaging of HwTxIV-Nvoc restores a fully functional HwTxIV.** UV illumination at 365 nm of HwTxIV-Nvoc fully produced uncaged HwTxIV with an uncaging half-time of 3.6 min (45 mW/cm$^2$) and half-power of 11.8 mW/cm$^2$ (Fig. 3a, b). HwTxIV released from uncaging of HwTxIV-Nvoc has the same elution time by RP-HPLC and molecular weight as synthetic noncaged HwTxIV (Fig. 3c, d). After illumination, the NMR spectra of uncaged HwTxIV and noncaged HwTxIV are perfectly superimposable indicating that illumination restores the predicted non-caged peptide (Fig. 3e). Also, photolysis of HwTxIV-Nvoc occurred from to λ = 365 nm up to 405 nm but was negligible at wavelengths in the 435–740 nm range (10 min, power> 18 mW/cm$^2$) which offers the possibility to use this light-triggered uncaging technology in combination with fluorophores excited above 405 nm for additional monitoring techniques (Supplementary Fig. 4). Complete uncaging of HwTxIV-Nvoc leads to inhibitory properties on hNa$_V$1.6 that are identical to noncaged toxin, confirming structure preservation upon uncaging, but also indicating that the addition of the Nvoc protecting

group did not lead to an abnormal disulphide bridging during peptide synthesis (Fig. 3f, g). As expected from the various concentration-response curves measured so far, illumination of 100 nM HwTxIV-Nvoc that is by itself inactive at this concentration, leads to a significant inhibition of hNa$_V$1.6 currents (Fig. 4a–c). Ending the illumination period does not restore hNa$_V$1.6 current levels, as Nvoc deprotection is irreversible (Fig. 4b). Similar light-induced block of Na$_V$ currents were observed with hNa$_V$1.2 (Fig. 4d, e). Also, as expected from a lipophilic toxin, HwTxIV blocked hNa$_V$1.6 in a poorly reversible manner after partial uncaging of 100 nM HwTxIV-Nvoc. Indeed, 11 min peptide washing was required to observe 16% of current recovery (n = 6; Supplementary Fig. 5). We also demonstrated that the extent of hNa$_V$1.6 inhibition increases as a function of illumination time and power which is expected by the progressively larger extent of uncaging occurring with illumination (Fig. 4f, g). As negative control, we show that illumination alone (365 nm, 250 s, 45 mW/cm$^2$) has no impact on Na$^+$ current amplitude (Supplementary Fig. 6a). Moreover, the Nvoc photolysis by-product resulting from uncaging of 100 nM BeKm1-Nvoc toxin, used as a negative control since BeKm-1 does not act on Na$_V$ channels, is inert on hNa$_V$1.6 (Supplementary Fig. 6b).

**Generalisation of caging approach for toxins modulating ion channels.** Similar inhibitory toxins displaying an important amine function on a key residue for pharmacology such as BeKm1 (hERG blocker)[40] and charybdotoxin (K$_V$1.2 blocker)[41] were also caged with a Nvoc protecting group (on K$^{18}$ for BeKm1 and on K$^{27}$ for charybdotoxin based on previous SAR studies[40,41]). In addition, a Nvoc-grafted analogue of AaHII (a Na$_V$ channel activator) was also produced wherein Arg$^{62}$, another essential residue for function[8], was replaced by Lys$^{62}$ with a Nvoc protecting group to enlarge the applicability of the technique. Those toxins present similar uncaging efficacies, indicating that cage removal is largely toxin amino acid sequence- and conformation-independent (Supplementary Fig. 7). These results suggest that this peptide toxin caging/uncaging approach is widely applicable regardless of amino acid sequences. As expected for an appropriate positioning of Nvoc on the toxin sequence, uncaging by illumination at 365 nm of AaHII-R$^{62}$K-Nvoc, BeKm1-Nvoc, or charybdotoxin-Nvoc lead to UV-dependent effects on hNa$_V$1.2 (activation), hERG (block), and K$_V$1.2 (block) currents (Fig. 4h), respectively. This result indicates that this approach can be generalized to toxin activators and pore blockers for the control of a wide diversity of ion channels.

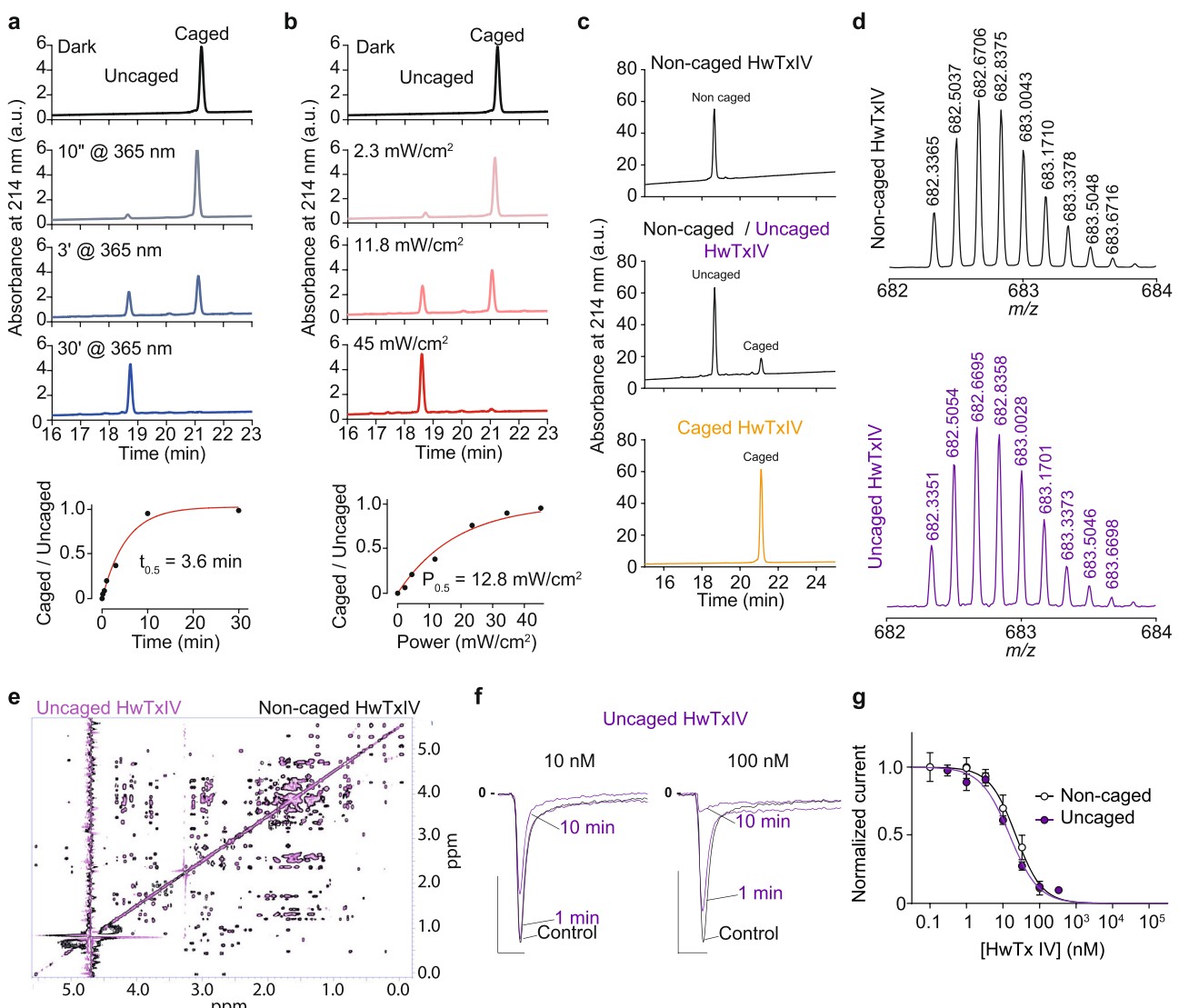

**Fig. 3 Physico-chemical and electrophysiogical properties of HwTxIV-Nvoc. a** Top: Analytical RP-HPLC profiles of HwTxIV-Nvoc after different durations of illuminations (365 nm, 45 mW/cm²) demonstrating time-dependent control of uncaging. Bottom: Uncaged/Caged ratio of HPLC chromatogram peaks area versus irradiation time at 365 nm (45 mW/cm²). **b** Top: Analytical RP-HPLC profiles of HwTxIV-Nvoc after different intensities of illuminations (365 nm, 5 min) demonstrating intensity-dependent control of uncaging. Bottom: Uncaged/Caged ratio of HPLC chromatogram peaks area versus irradiation power at 365 nm (3 min). **c** Analytical RP-HPLC profiles of non-caged HwTxIV (top), 50:50 ratio of noncaged HwTxIV after partial uncaging of HwTxIV-Nvoc (middle) and of HwTxIV-Nvoc (bottom). **d** Mass analyses of noncaged HwTxIV and uncaged HwTxIV by LC-ESI QTOF ([M + 6H]⁶⁺ $[M + 6H]^{6+}$ values). **e** Superimposition of TOCSY spectra (same area as Fig. 1e) of the noncaged HwTxIV (in black), and of the uncaged HwTxIV after illumination of HwTxIV-Nvoc (in purple). **f** Representative recording of hNa$_V$1.6 currents elicited at 10 mV from a holding potential of -100 mV illustrating the extent of current block by different concentrations of non-caged HwTxIV (black) and uncaged HwTxIV (purple). Scales: 2 ms, 1 nA. **g** Average dose-response curves for hNa$_V$1.6 current inhibitions by non-caged HwTxIV (black) and uncaged HwTxIV (purple). The data were fitted according to a Hill equation (noncaged HwTxIV IC$_{50}$ = 22.4 nM ($n$ = 5) and uncaged HwTxIV IC$_{50}$ = 15.20 nM ($n$ = 6)). Data are presented as mean ± SEM. Source data are provided as a Source Data File.

**Spatiotemporal control of HwTxIV activity in brain slices**. To expand our approach to biological systems, we assessed the use of HwTxIV-Nvoc in mouse brain slices during optical measurements of Na⁺ influx. We first determined that action potentials (APs) recorded in neocortical layer-5 (L5) pyramidal neurons were inhibited by local application of 500 nM noncaged HwTxIV from the surface of the brain slice near the cell body (Fig. 5a). Similar experiments conducted with HwTxIV-Nvoc show that AP shape is unaltered (Fig. 5b, c). Unsurprisingly, photolysis of HwTxIV-Nvoc leads to a significant decrease of the maximal membrane potential (V$_m$) 1 min after illumination (Fig. 5b, c). As expected from the time-course of reversal in the uncaging

experiments performed on the hNa$_V$1.6-expressing cell line (Supplementary Fig. 5), the effect of uncaged HwTxIV was poorly reversible in brain slices. It took 30 min to get a reasonable washout of the peptide toxin (Supplementary Fig. 8). Next, spatial selectivity was first examined using a ~100 μm diameter 365 nm UV LED spot. In this configuration, the somatic AP was not affected by illuminating a spot centered ~100 μm away from the soma, whereas direct illumination of the soma inhibited the somatic AP (Fig. 5d, e). We next used a more precise ~40 μm diameter illumination spot to define the spatial conditions for toxin uncaging/AP blocking by positioning the spot at varying distances from the soma (Fig. 5f, g). The somatic AP was not

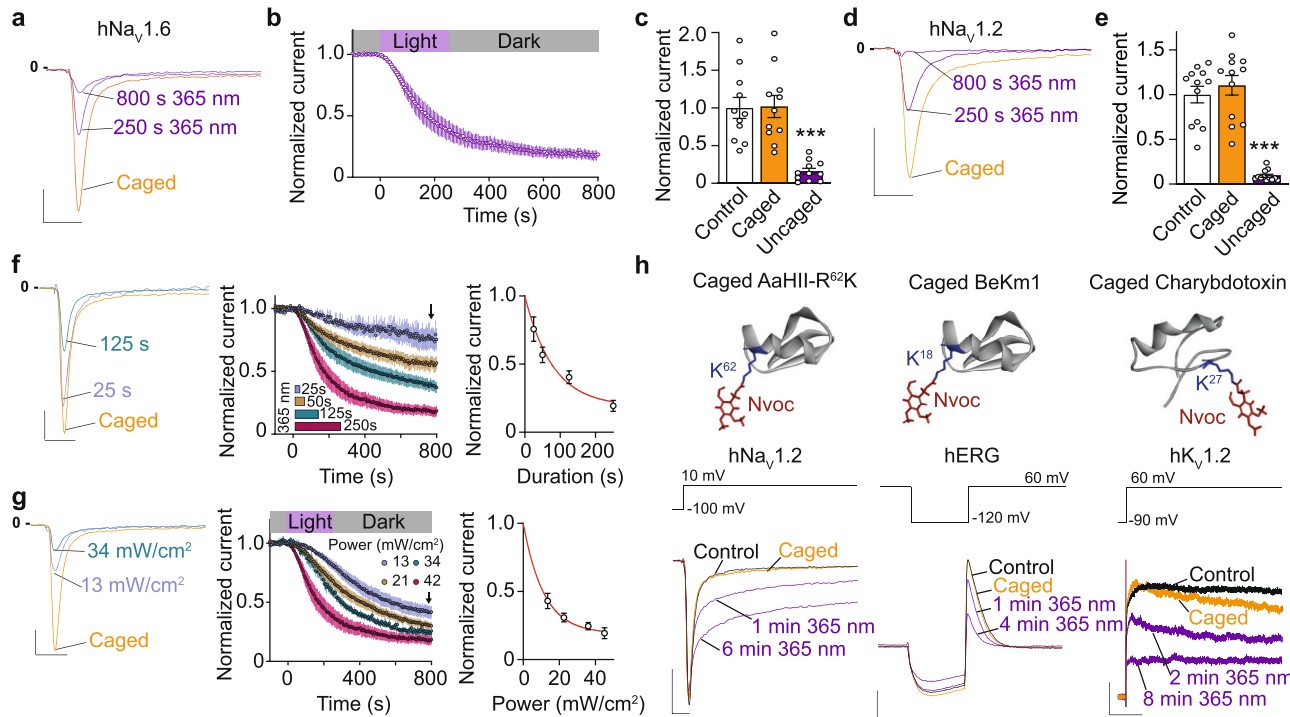

**Fig. 4 Modulation of ion channels properties by light-induced uncaging of peptide toxins. a–c** Light-induced inhibition of hNa$_V$1.6 current by photolysis of 100 nM HwTxIV-Nvoc. Note that not all of the HwTxIV-Nvoc is uncaged by photolysis during this exposure time. **a** Representative recordings of hNa$_V$1.6 current with caged HwTxIV (orange) and after various illumination times (purple) and (**b**) average normalized time courses of hNa$_V$1.6 current inhibition during and following light application ($n = 11$, mean ± SEM). Scale: 2 ms, 1 nA. **c** Average normalized current at steady-state in control (dark), caged (orange) and uncaged (purple) conditions ($n = 11$, ***$p < 0.001$, two-sided repeated-measures 1-way ANOVA followed by Bonferroni's post-test.). **d** Representative recordings of hNa$_V$1.2 current with caged HwTxIV (orange) and after illumination (purple). Scale: 2 ms, 1 nA. **e** Average normalized current at steady-state in control (dark), caged (orange), and uncaged (purple) conditions ($n = 12$, mean ± SEM ***$p < 0.001$, two-sided repeated-measures 1-way ANOVA followed by Bonferroni's post-test). **f–g** Controllable uncaging of HwTxIV-Nvoc via UV irradiation. **f** Left: Superimposition of normalized hNa$_V$1.6 currents at steady-state recorded after varying durations of illumination. Middle: Average normalized time courses of hNa$_V$1.6 current inhibition ($n = 40$ cells, mean ± SEM). Arrow indicates steady-state. Right: Plot of current amplitude at the end of 800 s recording versus duration of illumination at 365 nm. Scale: 2 ms, 20% of amplitude. **g** Left: Superimposition of normalized hNa$_V$1.6 currents at steady-state recorded after varying power of illumination (4 min). Middle: Average normalized time courses of hNa$_V$1.6 current inhibition ($n = 43$ cells, mean ± SEM). Arrow indicates steady-state. Right: Plot of current amplitude at the end of 800 s recordings versus duration of illumination at 365 nm. Scale: 2 ms, 20% of amplitude. **h** Representative examples of photocontrol (365 nm, 45 mW/cm²) of 1 nM AaHII-R$^{62}$K-Nvoc activity on hNa$_V$1.2 current (left, AaHII PDB code 6NT4[8]), 100 nM BeKm1-Nvoc activity on hERG current (middle, BeKm-1 PDB code 1J5J[40]) and 100 nM charybdotoxin-Nvoc activity on K$_V$1.2 current (right, ChTx PDB code 4JTA[41]). AaHII-R$^{62}$K toxin induces slowing of inactivation of hNa$_V$1.2, while BeKm1 and charybdotoxin induce block of hERG and K$_V$1.2 channels, respectively. For AaHII-R$^{62}$K-Nvoc and BeKm1-Nvoc, scale is 2 ms and 1 nA. For charybdotoxin-Nvoc, scale is 400 ms and 200 pA. Source data are provided as a Source Data File.

affected when the cell was >80 µm from the spot center, whereas it was partially inhibited at distances between 60 and 20 µm from the spot center (Fig. 5h). These results provide information about the spatial resolution that can be attained by uncaging of HwTxIV-Nvoc inside the area of photolysis. We then recorded Na$^+$ influx via Na$_V$ channels, associated with an AP, to unambiguously assess the effect of the uncaged toxin on the channels expressed in the axon initial segment (AIS) (Fig. 5i) using an ultrafast Na$^+$ imaging approach[42]. Illumination of HwTxIV-Nvoc in the soma (500 ms, 2 mW) fully prevented Na$^+$ influx in the AIS, even at the same depolarized V$_m$ of the AP (Fig. 5j, k, Supplementary Movie 1). This result demonstrates that the photo-release of HwTxIV blocks Na$_V$ channels and AP propagation even at positive V$_m$. Altogether, these results provide proof-of-principle that controlled photolysis of HwTxIV-Nvoc can be used to precisely target AP initiation and Na$^+$ influx in the AIS in brain slices.

**In vivo spatiotemporal control of caged toxins.** In order to evaluate the functional in vivo effects of our approach, we first

validated that intraperitoneal injection of HwTxIV-Nvoc has negligible effects on mice activity contrary to a 10-fold lower amount of noncaged HwTxIV which drastically reduces motor behavior (Fig. 6a–c). Using in situ EDL muscle contraction monitoring in anesthetized mice, we validated that muscle illumination at 365 nm has no effects on EDL twitches by itself in control mice but demonstrated that a similar illumination paradigm of mice injected with HwTxIV-Nvoc (365 nm, 50 mW/cm²) significantly reduces contractile force thanks to uncaging of the peptide at the neuromuscular junctions where Na$_V$1.6 is expressed (Fig. 6d–e). This result demonstrates in vivo activation of caged peptide toxins using UV light in superficial tissues. The data also indicate that the caged peptide reaches concentrations at the neuromuscular junctions high enough for light activation in spite of a simple intraperitoneal injection. By using a zebrafish larvae model, we confirmed the efficacy of this approach using AaHII-R$^{62}$K-Nvoc peptide. Indeed, zebrafish illumination induces a significant reduction of swimming movements and a paralysis in all (6/6) larvae previously injected with 50 µM AaHII-R$^{62}$K-Nvoc (Supplementary Fig. 9 and Supplementary Movies 2 and 3). Illumination of control larvae in the same conditions

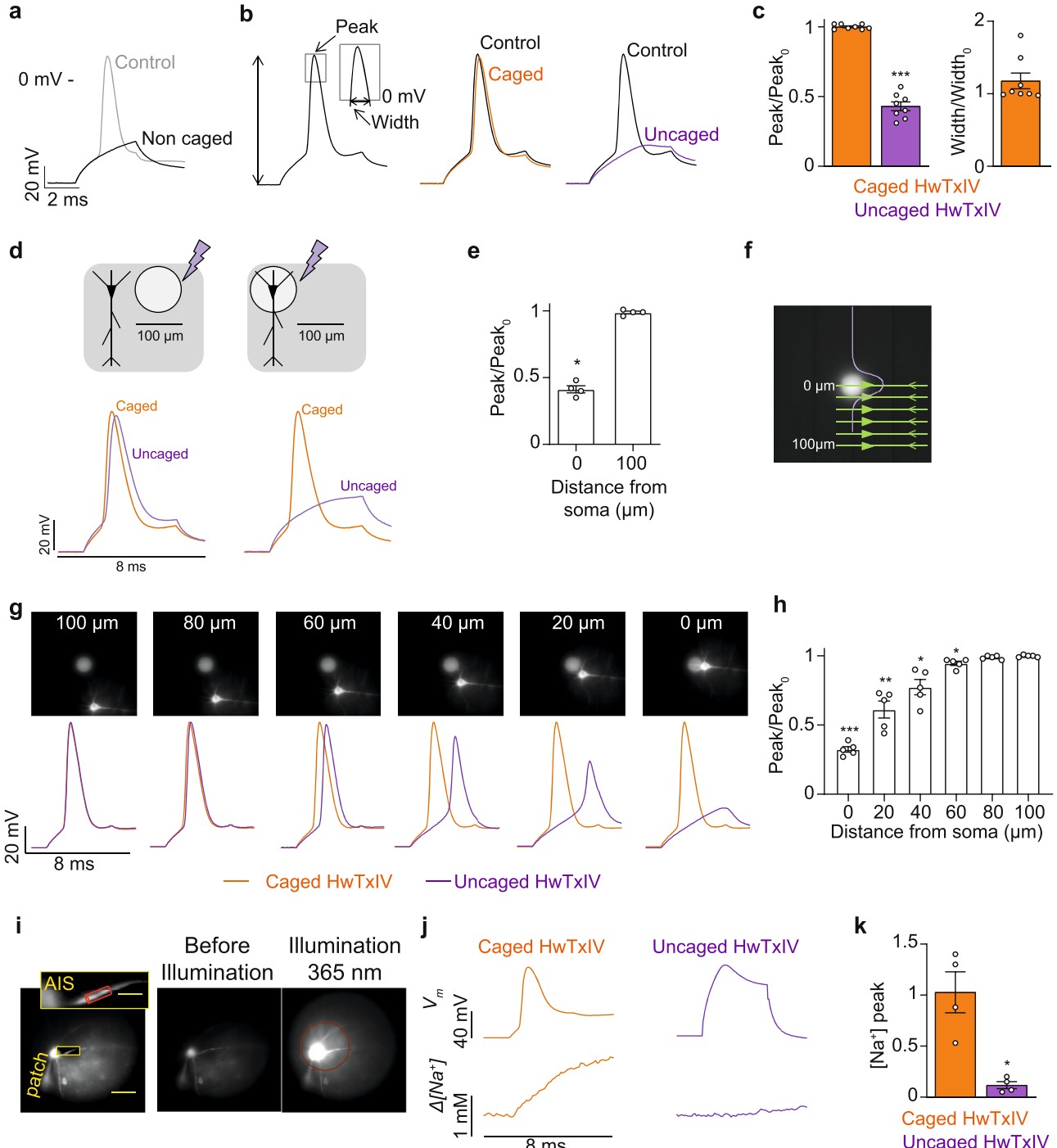

**Fig. 5 Control of brain slice L5 pyramidal neurons by HwTxIV-Nvoc. a** Inhibition of AP by 500 nM non-caged HwTxIV ($n = 6$). **b** AP in control condition, in presence of 2.5 µM HwTxIV-Nvoc and 1 min after uncaging. **c** Left: mean ± SEM ($n = 8$ cells) of normalized AP peak after addition of HwTxIV-Nvoc and 1 min after photolysis (*$p = 0.0078$, two-sided paired $t$ test). Right: mean ± SEM ($n = 5$ cells) of normalized AP width after addition of HwTxIV-Nvoc. **d** APs 1 min after HwTxIV-Nvoc uncaging over a spot centered ~100 µm from the cell body or over the soma. **e** Mean ± SEM ($n = 4$ cells) of the normalized AP peak after uncaging 100 µm away from the soma or directly on the soma. (*$p = 0.0005$, two-sided paired $t$ test). **f** Image of the ~40 µm diameter UV spot. Purple curve represents the light intensity profile on cell schematic. **g** Top: images of a L5 pyramidal neuron relative to UV (405 nm) illumination spot position. Bottom: AP in the presence of HwTxIV-Nvoc (orange traces) and 1 min after uncaging (purple traces) with cell positioned at 100, 80, 60, 40, 20, and 0 µm distance from spot. **h** Mean ± SEM ($n = 5$ cells) of normalized AP peak 1 min after uncaging with the cell positioned at decreasing distances from spot center (100 µm; 80 µm, $p = 0.2302$; 60 µm, *$p = 0.0434$; 40 µm, *$p = 0.0420$; 20 µm, *$p = 0.0133$; 0 µm, ***$p < 0.0001$). Two-sided paired $t$-test. **i** Left, L5 pyramidal neuron filled with 500 µM of ING-2. AIS area of $\Delta[Na^+]$ measurement in red cylinder. Right, images before and during UV uncaging pulse illustrating photolysis area. **j** Left, somatic AP (top) and associated $\Delta[Na^+]$ signal in the presence of HwTxIV-Nvoc. Right, after toxin uncaging, the cell was depolarized to +20 mV to correspond to control AP peak and measure $\Delta[Na^+]$ signal in the same condition. **k** mean ± SEM ($n = 4$ cells) of the $\Delta[Na^+]$ signal maximum (peak) before and after uncaging the toxin. (*$p < 0.0174$, two-sided paired $t$-test). Source data are provided as a Source Data File.

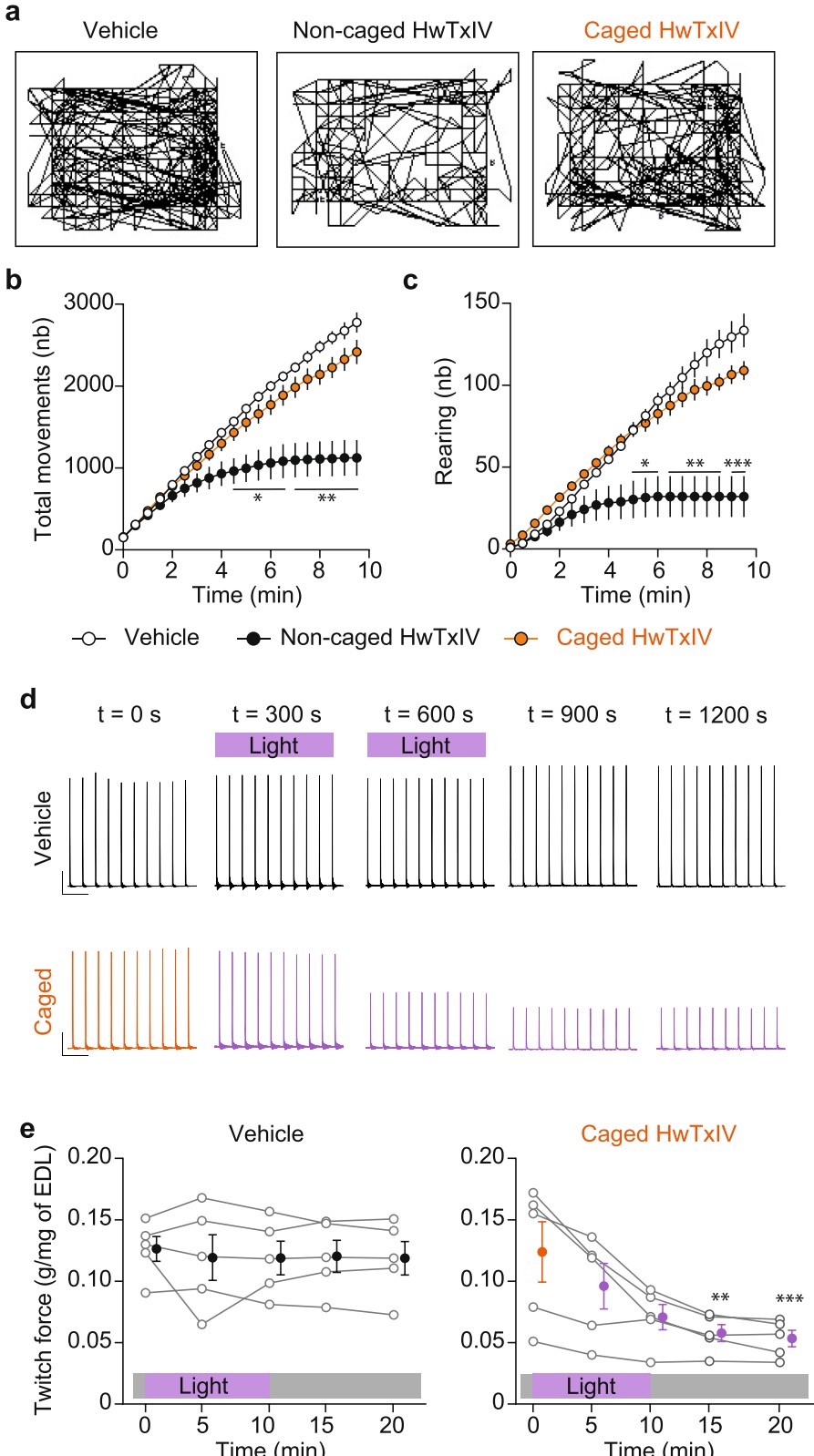

**Fig. 6 In vivo uncaging of HwTxIV-Nvoc. a** Representative trajectory plots of mice injected with vehicle (left), non-caged HwTxIV (center) or HwTxIV-Nvoc (right). **b–c** Quantification of total movements (**b**) and rearing (**c**) in mice injected with vehicle (white circles), HwTxIV (black circles) or HwTxIV-Nvoc (orange circles). ($n = 5$ in each group, mean ± SEM *$p < 0.05$; **$p < 0.01$; ***$p < 0.001$ versus vehicle, two-sided repeated-measures 2-way ANOVA test followed by Tukey's post test). **d** Representative twitches obtained from mice injected with vehicle or HwTxIV-Nvoc at $t = 0$; $t = 5$; $t = 10$; $t = 15$; $t = 20$ min after injection. **e** EDL twitch force normalized to muscle mass (g/mg of EDL) in vehicle or HwTxIV-Nvoc mice before and after illumination (365 nm, 50 mW/cm$^2$). ($n = 5$ in each group, mean ± SEM **$p = 0.0041$; ***$p = 0.0002$ versus t0, two-sided repeated-measures Friedman test followed by Dunn's post-test). Source data are provided as a Source Data File.

induce a nonsignificant reduction of movements largely due to habituation to the new environment in which the larvae were positioned. Those in vivo approaches demonstrate the potential of caged peptide toxins to be ineffective prior to illumination but fully active in spatially restricted regions after illumination in order to control ion channels in an integrated model.

## Discussion

In summary, we report the development and application of a robust, generalizable, and in vivo compatible strategy for producing photoactivatable peptide toxins modulating voltage-gated ion channels and cell excitability. By using HwTxIV-Nvoc, we demonstrate that caged compounds can be activated by wavelengths <435 nm and is therefore compatible with dyes used for optical imaging or fluorescent compounds to monitor voltage-dependent structural changes in ion channels[43]. It is worth noting that this approach allows a spatial and temporal control of voltage-gated ion channel function. Our findings provide opportunities for gaining insights into the functional role of ion channels in physiological or pathological processes such as sensory perception disorders, muscle and brain channelopathies[44] or cardiac muscle diseases[45]. The utility of classical pharmacology in vivo is limited, because local drug delivery is temporally ill-defined and imprecise. Often also, the far-from-perfect selectivity profile of peptide toxins leads to unwanted side effects in vivo. In contrast, photopharmacology is able to mimic the timing, amplitude, and spread of naturally occurring modulatory signals. By offering a real-time control of voltage-gated ion channels complexes and their functionalities in precisely defined regions that are poorly accessible to electrophysiological or pharmacological manipulations, photoactivatable peptide toxins open a window for the understanding of the physiology of ion channels. With the methodology described herein, the need for selective compounds is also attenuated because a channel subtype of interest can be targeted selectively thanks to the precision of spatial uncaging. In the case of HwTxIV, acting like a pan-$Na_V$ inhibitor for $Na_V1.1$, $Na_V1.2$, $Na_V1.6$, and $Na_V1.7$, the selectivity of action on $Na_V1.6$ is provided by uncaging the compound at the neuromuscular junction, within a tissue, the skeletal muscle, where $Na_V1.6$ predominates but lacking the three other channel subtypes. Similar approaches have been previously reported for ligand-gated ion channels[46] and an elegant approach using coumarin-derived protected group grafted on saxitoxin targeting sodium channels has been recently published[17]. Despite very fast activation, this approach has been used on a nonpeptidic toxin and proof of in vivo efficacy has not been demonstrated yet. In our study, we demonstrate the ability of the caged peptide to be used, in vivo, as light-sensitive modulators of ion channels both in mice and zebrafish larvae. Given the number of peptides being isolated from venoms that target membrane proteins, the use of these photosensitive groups will first enable the development of a large number of additional photoactivable toxins to better understand the functional heterogeneity of ion channels and next, by establishing causal relationships between a protein activity and a cellular or physiological output, photoactivatable peptide toxins hold strong potential for identifying new therapeutic targets in many ion channel-related diseases. Peptide toxins of venom origin seem to hold the greatest promises as they have been tailored by nature for proper pharmacological action in vivo (adequate distribution and stability) and display a variety of properties in terms of selectivity and reversibility. In our case, low reversibility of HwTxIV is a desired property for prolonged muscle paralysis or chronic pain treatment for instance. In brain tissues, where experimentalists like to repeat the observation from multiple cells, more reversible peptide toxins than HwTxIV would be preferred

for rapid clearance of the effect. The main limitation that can be invoked is the fact that this technology is based on peptides, poorly delivered by oral ingestion, more expensive to be produced than small chemicals and with more limited stability in vivo, although considerable progress is witnessed lately in using peptides as therapeutics.

## Methods

**Ethical Statement**. All animal care and experimental procedures were performed in the animal facilities that have been accredited by the French Ministry of Agriculture. The experimental procedures were approved by the regional ethic committees (CEEA-006 Pays de la Loire; CEEA-012 Grenoble; CEEA-LR Languedoc Roussillon, France) according to the Directive 2010/63/EU of the European Union.

**Molecular modelling**. To understand if modifications in toxin activity are connected to alterations in peptide/channel interaction, molecular simulations were performed. The structures of $Na_VAb/Na_V1.7$-VS2A chimera with m3-Huwentoxin-IV (pdb code 7K48)[30] was used as a template for i) illustrating the docking of HwTxIV on $Na_V1.7$, ii) deciding that $K^{32}$ was a good candidate for caging, and iii) highlighting the clashes induced by the Nvoc presence on $K^{32}$. Discovery Studio (Dassault System) was used to build HwTxIV-Nvoc structure. After applying a fast, Dreiding-like forcefield in order to clean geometry of designed peptide, the van-der Waals (vdW) repulsion forces between HwTxIV-Nvoc and $Na_V1.7$ were calculated using the "Show bumps" plugin, implemented in PyMOL (http://www.pymolwiki.org/index.php/Show_bumps).

**Chemical syntheses of HwTxIV and peptide analogues**. All peptides were assembled stepwise using Fmoc-based Solid Phase Peptide Synthesis (SPPS) on a PTI Symphony synthesizer at a 0.05 mmol or 0.1 mmol scale on 2-chlorotrityl chloride polystyrene resin (substitution approximately 1.6 mmol/g). For caged peptides, Fmoc-L-Lysine residues at position 32 (HwTxIV), 18 (BeKm1), 27 (charybdotoxin), and 62 (AaHII-R$^{62}$K) were replaced by Fmoc-L-Lys(Nvoc)-OH (Iris Biotech, Marktredwitz, Germany) during assembly. The Fmoc protecting group was removed using 20% piperidine in DMF and free amine was coupled using tenfold excess of Fmoc amino acids and HCTU/DIEA activation in NMP/ DMF (3×15 min). Linear peptides were de-protected and cleaved from the resin with TFA/$H_2$O/1,3-dimethoxybenzene(DMB)/TIS/2,2′-(Ethylenedioxy)diethanethiol(DODT) 85.1/5/2.5/3.7/3.7 (vol.), then precipitated out in cold diethyl ether. The resulting white solids were washed twice with diethyl ether, re-suspended in $H_2$O/acetonitrile, and freeze dried to afford crude linear peptide. Oxidative folding of the crude linear HwTxIV-Nvoc was successfully conducted at RT in the conditions optimized for the HwTxIV using a peptide concentration of 0.1 mg/mL in a 0.1 M Tris buffer at pH 8.0 containing 10% of DMSO. Similar conditions were used for the syntheses of AaHII-R$^{62}$K-Nvoc, BeKm1-Nvoc, and charybdotoxin-Nvoc. Purification of the peptides were performed on Chromolith®HighResolution RP-18e column (100 ×4.6 mm) using a 5–65% buffer B run in 14 min at 2 mL/min. Solvent system: A: $H_2$O 0.1%TFA and B: MeCN 0.1%TFA.

**LC-ESI-QTOF MS analyses**. LC-ESI-MS data were acquired with an Accurate mass QTOF 6530 (Agilent) coupled to an Agilent 1290 UPLC system. If required, separation of the peptide sample (5 µL, approx. 10 µg/mL) was done on an Ascentis express 90 Å C18 column (2.0 µm, 2.1 mm ID × 100 mm L, Supelco) heated at 70 °C at a flow rate of 0.5 mL/min and with a 2-70% buffer B/A gradient over 11 min (buffer A: $H_2$O/formic acid, 99.9/0.1 (v/v); buffer B: ACN/formic acid, 99.9/ 0.1 (v/v)). If not required, the sample was directly injected at 0.6 mL/min flow rate in $H_2$O/MeCN +0.1% formic acid. The acquisition was done in the positive mode with a Dual ESI source, within a m/z 100-1700 mass range and analyzed via the Agilent MassHunter software version 10.0. Source parameters were set as follows: capillary voltage, 4 kV; fragmentor, 100 V; gas temperature, 350 °C; drying gas flow, 12 L/min; nebulizer 50 psig.

The observed monoisotopic masses of the peptides were as follow: AaHII-R$^{62}$K: 7210.21 Da (theoretical: 7210.14); AaHII-R$^{62}$K-Nvoc: 7449.23 (theoretical: 7449.18); BeKm-1: 4088.79 Da (theoretical 4088.78); BeKm-1-Nvoc: 4327.85 Da (theoretical: 4327.81); Charybdotoxin: 4292.84 Da (theoretical: 4292.87); Charybdotoxin-Nvoc: 4531.94 Da (theoretical: 4531.91); HwTxIV: 4088.04 Da (theoretical: 4087.96); HwTxIV-Nvoc: 4327.08 Da (theoretical: 4327.00). Source data are provided as a Source Data File.

**Monitoring of Nvoc deprotection by analytical RP-HPLC**. Analytical RP-HPLC was performed using an SPD M20-A system (Shimadzu) with a Luna OmegaPS C18 column (4.6 ×250 mm, 5 µm, 100 Å). Two µL (corresponding to 7 µg of material) was loaded and a 5-60% acetonitrile gradient (0.1% TFA v/v) was applied over 35 min at room temperature to detect analytes by UV absorbance at 214 nm. Illumination of samples was performed at 365 nm for different times (between 1 sec and 30 min illumination time) at 41.8 mW/cm$^2$ or less (as specified in the Result section) for 10 min using a CoolLED pE4000 light source (CoolLED, UK). Flash energy has been measured using a High Sensitive Thermal Power Head (S401C,

ThorLabs). Coelution of uncaged HwTxIV and non-caged HwTxIV was performed using a 50:50 ratio for both compounds.

**NMR spectrometry**. HwTxIV-Nvoc was illuminated for 30 min at 100% power (45 mW/cm$^2$) using a CoolLED pE4000 light source (CoolLED, UK). Three 200 µL solutions were used in 3 mm NMR tubes for i) HwTxIV-Nvoc before illumination (500 µM); ii) HwTxIV-Nvoc after illumination (500 µM); and iii) non-caged HwTxIV (200 µM). For each sample, two-dimensional homonuclear (80 ms TOCSY and 160 ms NOESY) and heteronuclear $^{13}C$-HSQC (natural abundance) spectra were acquired on a BRUKER 700 MHz NMR spectrometer equipped with a 5 mm TCI cryoprobe, at 298 K. Processing and analyses were performed with Bruker's TopSpin3.2 and CcpNMR programs. Spectra are drawn with CcpNMR program. 3D structure is drawn with the PyMOL software (The PyMOL Molecular Graphics System, Version 2.0 Schrödinger, LLC).

**Cell cultures**. HEK293 cells stably expressing the hNa$_V$1.1 (XM_011511604), hNa$_V$1.2 (XM_017004655) or hNa$_V$1.6 (XM_011538651) channels, CHO cells transiently expressing hK$_V$1.2 (NG_027997.2) and CHO cells stably expressing hNa$_V$1.7 (XM_011511618) or the hERG (NM_000238) channels were cultured in Dulbecco's Modified Eagle's Medium (DMEM) supplemented with 10% fetal bovine serum, 1 mM pyruvic acid, 4.5 g/L glucose, 4 mM glutamine, 800 µg/mL G418, 10 U/mL penicillin and 10 µg/mL streptomycin (Gibco, Grand Island, NY). All cell lines were incubated at 37 °C in a 5% CO$_2$ atmosphere. For electrophysiological recordings, cells were detached with trypsin and floating single cells were diluted (~300,000 cells/mL) in medium contained (in mM): 140 mM, 4 KCl, 2 CaCl$_2$, 1 MgCl$_2$, 5 glucose, and 10 HEPES (pH 7.4, osmolarity 298 mOsm).

**Automated patch-clamp recordings and pharmacological studies**. Whole-cell recordings were used to investigate the effects of HwTxIV peptides on HEK293 or CHO cells expressing hNa$_V$1.1, hNa$_V$1.2, hNa$_V$1.6, or hNa$_V$1.7 channels but also the effects of AaHII-R$^{62}$K-Nvoc, BeKm1-Nvoc, and charybdotoxin-Nvoc on hNa$_V$1.2, hERG, and hK$_V$1.2, respectively. Automated patch-clamp recordings were performed using the SyncroPatch 384PE from Nanion (München, Germany). Chips with single-hole medium resistance of $4.52 \pm 0.08$ MΩ ($n = 384$) were used for recordings. Pulse generation and data collection were performed with PatchControl384 v1.5.2 software (Nanion) and the Biomek v1.0 interface (Beckman Coulter). Whole-cell recordings were conducted according to the recommended procedures of Nanion. Cells were stored in a cell hotel reservoir at 10 °C with shaking speed at 60 RPM. After initiating the experiment, cell catching, sealing, whole-cell formation, liquid application, recording, and data acquisition were all performed sequentially and automatically. For sodium channel recordings, intracellular solution was (in mM): 10 CsCl, 110 CsF, 10 NaCl, 10 EGTA, and 10 HEPES (pH 7.2, osmolarity 280 mOsm), while being 10 KCl, 10 KF, 10 NaCl, 10 EGTA, and 10 HEPES (pH 7.2, osmolarity 280 mOsm) for potassium channel recordings. Extracellular solution was (in mM): 140 NaCl, 4 KCl, 2 CaCl$_2$, 1 MgCl$_2$, 5 glucose, and 10 HEPES (pH 7.4, osmolarity 298 mOsm). For sodium channel recordings at room temperature (18–22 °C), holding potential was set at −100 mV and the sampling rate set at 20 kHz. Peptides were diluted in the extracellular solution containing 0.3% bovine serum albumin (BSA). Compounds were tested at 0 mV with 50 ms test potentials repeated every 5 s. Current inhibition percentages were measured at equilibrium after 14 min peptide application time. For hERG current, whole-cell experiments were performed at a holding potential of −80 mV and BeKm1-Nvoc was tested at a 200 ms test potential of +60 mV following a first activation step of 1000 ms at +60 mV and a 10 ms step at −120 to recover from inactivation with a pulse every 8 s. For K$_V$1.2 current, whole-cell experiments were performed at a holding potential of −90 mV and charybdotoxin-Nvoc was tested at a 2000 ms test potential of +60 mV with a pulse every 12 s.

**Photoactivation of HwTxIV-Nvoc on HEK293 cells**. The pharmacology of HwTxIV-Nvoc, as well as the efficiency of the released product to block hNa$_V$ channels was studied using combined automated patch-clamp and UV illumination. After 2 min of control, HwTxIV-Nvoc was added to the external buffer and effects were recorded for 100 s prior to a 250 s-duration illumination for photocleavage induction of the caged compound. Different wavelengths, illumination powers' and durations were used as specified in figure legends. For AaHII-R$^{62}$K-Nvoc, BeKm1-Nvoc, and charybdotoxin-Nvoc, a 250 s-duration illumination at 45 mW/cm$^2$ was used to uncage compounds.

**Construction of hNa$_V$1.6/rK$_V$2.1 chimeras**. Channel chimeras were generated using sequential PCR with K$_V$2.1Δ7[47,48] (Genscript, USA) and hNa$_V$1.6 (NM_014191, Origene Technologies, USA) as templates. cRNA was synthesized using T7 polymerase (mMessage mMachine kit, Thermo Fisher, USA) after linearizing cDNA with appropriate restriction enzymes. This chimeric approach was previously shown to robustly indicate the binding locus of toxins[32]. Boundaries of the chimeras were chosen as reported previously in Fig. S2 of Bosmans et al. 2011[33] and include the transfer of hNa$_V$1.6 regions (MMAYITEFVNLGNVSALRTFRVLR (VSDI), GFIVSLSLMELSLADVEGLSVLRSFRLLR (VSDII), SLVSLIANALGY-SELGAIKSLRTLRALR (VSDIII), SIVGMFLADIIEKYFVSPTLFRVIRLARIGRILR (VSDIV) into K$_V$2.1.

**Two-electrode voltage-clamp recordings**. Two-electrode voltage-clamp recording techniques on *Xenopus laevis* oocytes (OC-725C, Warner Instruments, USA; 150 µL recording chamber) were used to measure channel currents 1 day after cRNA injection and incubation at 17 °C in ND96 that contained (in mM): 96 NaCl, 2 KCl, 5 HEPES, 1 MgCl$_2$, and 1.8 CaCl$_2$, 50 µg/mL gentamycin, pH 7.6. Data were filtered at 4 kHz and digitized at 20 kHz using pClamp software (Molecular Devices, USA). Microelectrode resistances were 0.5–1 MΩ when filled with 3 M KCl. For K$_V$ channel experiments, the external recording solution contained (in mM): 50 KCl, 50 NaCl, 5 HEPES, 1 MgCl$_2$ and 0.3 CaCl$_2$, pH 7.6 with NaOH. All experiments were performed at RT (~21 °C) and toxin samples were diluted in recording solution with 0.1% BSA. Voltage-activation relationships were obtained by measuring tail currents for K$_V$ channels. After the addition of toxin to the recording chamber, equilibration between toxin and channel was monitored using weak depolarizations elicited at 5–10 s intervals. For all channels, voltage-activation relationships were recorded in the absence and presence of the toxin. Off-line data analysis was performed using Clampfit 11 (Molecular Devices, USA) and Origin 8 (Originlab, USA).

**Mice**. For coronal slices, procedures were reviewed by the ethics committee affiliated to the animal facility of the university (D3842110001) and performed in accordance with European Directives 2010/63/UE on the care, welfare, and treatment of animals. Specifically, C57BL/6 J mice were housed with their mother with ad libitum access to food and water. Animals (21–35 postnatal days old) were anesthetised by isoflurane inhalation and the entire brain was removed after decapitation. For neuromuscular experiments, procedures were carried out on 25 male mice (C57BL/6 J mice) aged between 6 and 8 weeks. Mice were housed five per cage and maintained on a 12/12 h light/dark schedule in a temperature-controlled facility (22 ± 1 °C) with free access to food and water. Animals were kept undisturbed for 7 days before experiments. Animals were divided into five groups. For contractile in situ experiments, two groups of 5 mice received or a single dose of HwTxIV-Nvoc (5 mg/kg in 100 µL) or a similar intraperitoneally injection of 0.9% NaCl solution. For behavior actimeter experiments, three groups of five mice received a single intraperitoneally injection of HwTxIV (0.5 mg/kg in 100 µL) or HwTxIV-Nvoc (5 mg/kg in 100 µL) or similar intraperitoneally injection of 0.9% NaCl solution. All procedures were conducted in conformity with European rules for animal experimentation (French Ethical Committee APAFIS#8186-2016121315485337, January 10, 2017, for EDL contractile in situ experiments, APAFIS#1765-2016121315485337, January 10, 2017, for motor behavior experiments with native toxin).

**Electrophysiological recordings from neocortical brain slices**. Neocortical coronal slices (350 µm thick) were prepared and maintained using previously described procedures adapted from other preparations[49–52]. Briefly, slices were incubated in extracellular solution at 37 °C for 45 min and then maintained at room temperature before use. The extracellular solution contained (in mM): 125 NaCl, 26 NaHCO$_3$, 1 MgSO$_4$, 3 KCl, 1 NaH$_2$PO$_4$, 2 CaCl$_2$, and 20 glucose, bubbled with 95% O$_2$ and 5% CO$_2$. The intracellular solution contained (in mM): 125 KMeSO$_4$, 5 KCl, 8 MgSO$_4$, 5 Na$_2$-ATP, 0.3 Tris-GTP, 12 Tris-Phosphocreatine, 20 HEPES, adjusted to pH 7.35 with KOH. Layer-5 (L5) pyramidal neurons from the somatosensory cortex were selected and patched in a whole-cell configuration. Somatic action potentials (APs) were elicited by injecting current pulses of 3–5 ms duration and of 1–2 nA amplitude. In Na$^+$ imaging experiments, after the photo-release of the toxin, the current intensity was increased to 5–10 nA in order to depolarize the cell to the same V$_m$ corresponding to the AP peak in the control condition. The measured V$_m$ was corrected for the junction and the bridge potentials. The caged toxin was dissolved in the extracellular solution at 2.5 µM concentration and locally delivered either using a SmartSquirt micro-perfusion system (WPI, Hitchin, UK) with a tip of 250 µm diameter, or by simple pressure ejection with a tip of ~10 µm. Photolysis of the caged compound was performed either on a spot of ~100 µm diameter and using the light of a 365 nm LED (~2 mW of power) controlled by an OptoLED (Cairn Research, Faversham, UK), or on a spot of ~40 µm using a 300 mW / 405 nm diode laser (Cairn Research). The protocol at 365 nm consisted of 1–3 pulses of 100–300 ms duration and 1 s interval, or by a single pulse of 500 ms duration. The protocol at 405 nm consisted of 2 pulses of 500 ms duration and 5 s interval. In the case of simultaneous fluorescence recordings and UV stimulation, uncaging was optically combined to imaging by coupling the two excitation wavelengths similarly to what was done for an equivalent application[53].

**Na$^+$ imaging**. Optical measurements of [Na$^+$] from the AIS were obtained as previously described[42]. Briefly, neurons were loaded with 500 µM of the Na$^+$ indicator ING-2 (IonBiosciences, San Marcos, TX, USA) for 20–30 min after establishing the whole-cell configuration. Fluorescence was excited using a 520-mW line of a LaserBank (Cairn Research) band-pass filtered at 517 ± 10 nm, directed to the preparation using a 538 nm long-pass dichroic mirror. Emitted fluorescence was band-pass filtered at 559 ± 17 nm before being recorded with a DaVinci 2 K CMOS camera (SciMeasure, Decatur, GA) at 10 kHz with a pixel resolution of 30×128. Optical data, obtained by averaging 4 trials under the same condition, were corrected for bleaching using a trial without current injection. The fractional change of Na$^+$ fluorescence was expressed in terms of intracellular Na$^+$

concentration change ($\Delta[Na^+]$) using the previously published calibration for which 1% corresponds to 0.175 mM[42].

**Actimeter.** Mice were randomly assigned to 3 groups according to peptide toxin treatment as described above. The motor behavior was examined with an open field actimeter[54]. For this analysis, mice were individually placed in an automated photocell activity chamber (Letica model LE 8811, Bioseb, France) which consists of a plexiglass chamber (20 cm×24 cm×14 cm) surrounded by two rows of infrared photobeams. The first row of sensors was raised at a height of 2 cm for measuring the horizontal activity and the second row placed above the animal for vertical activity. Just after treatment administration via intraperitoneally injection, the spontaneous motor activity was measured for 10 min using a movement analysis system (Bioseb, France), which dissociates total movements and rearing (numbers). All the experiments were realized in a dark room.

**Contractile properties of fast-EDL muscle using sciatic nerve stimulation.** EDL muscle force properties were analyzed by using in situ muscle contraction measurements[55,56]. Briefly, mice were anesthetized by intraperitoneally injection of xylazine/ ketamine (10/100 mg/kg), and the adequacy of the anesthesia was monitored throughout the experiment. The skin was then carefully removed from the left leg, the sciatic nerve was carefully isolated and EDL muscle was dissected free with its blood supply intact. The foot and the tibia of the mice were fixed by two clamps, and the distal tendon of the EDL muscle was attached to a force transducer and positioned parallel to the tibia. Throughout the experimental procedure, the mice were kept on a heating pad to maintain normal body temperature, and the muscles were continuously perfused with Ringer solution. Stimulation electrodes were positioned at the level of the sciatic nerve, and connected to a pulse generator with stimulation characteristics of 0.2 ms duration, 6 V stimulation amplitude and 1 Hz frequency. The muscle was stretched before stimulation voltage was applied to produce the most powerful twitch contractions. Stimulation was maintained during all the experiments. Twitch parameters were measured prior to UV illumination, after 5 and 10 min of UV illumination (365 nm, 50 mW/cm²) and 5 and 10 min after UV illumination. At the end of the experiments, EDL muscles were rapidly dissected and weighted. Twitch forces were normalized in grams per milligram of fresh EDL muscles. All force data are expressed as percentage of the initial force, i.e. before illumination. All the experiments were realized in a dark room. Data were collected and stored for analyses with Chart v4.2.3 (PowerLab 4/25 ADInstrument, PHYMEP France).

**Zebrafish larvaes movement tracking.** Zebrafish larvaes (*Danio rerio*, wild type AB strain) were maintained under standardized conditions and experiments were conducted in accordance with local approval (APAFIS#4054-2016021116464098 v5) and the European Communities council directive 2010/63/EU. 120 h post fertilization (hpf) larvae were anaesthetized using tricaine methanesulfonate (MS222, Sigma-Aldrich). Larvae were injected (microinjector-World Precision Instruments) in the pericardium with 1 nL of either vehicle (Texas Red) or a 50 µM solution of AaHII-$R^{62}$K-Nvoc. Locomotor activity was recorded in a dark environment inside a DanioVision observation chamber coupled with Ethovision video tracking v. 14 (Ethovision XT, Noldus Information Technology, Netherlands). After 10 min of tracking, larvae were exposed to 365 nm light (Prismatix) for 5 min. After a period of 30 min locomotor activity was recorded as before.

**Statistics and data analyses.** Values are represented as mean ± SEM. The significance of illumination tests on automated patch-clamp was tested by performing paired 1-way ANOVA and Bonferroni's multiple and significance of in vivo experiments was tested by performing Friedman test with Dunn's multiple comparisons test. A p-value lower than 0.05 was considered significant. Significance of pharmacological tests on electrophysiological and optical recordings in brain slices was tested by performing the paired t-test with 0.01 as probability threshold value.

**Reporting Summary.** Further information on research design is available in the Nature Research Reporting Summary linked to this article.

## Data availability
Data regarding chemical syntheses and electrophysiology generated in this study are provided in the Source Data File. HwTxIV BMRB and PDB codes used for NMR experiments are respectively 5527 and 1MB6. Source data are provided with this paper.

## Code availability
R codes used for electrophysiological analyses and Matlab code related to sodium imaging in brain slices are available on GitHub (https://github.com/jerome-montnach/Caged-HwTxIV.git).

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

## Acknowledgements

M. De Waard and M. Canepari thank the Agence Nationale de la Recherche (ANR) for financial support to the laboratory of excellence "Ion Channels, Science and Therapeutics" (grant N° ANR-11-LABX-0015). Both researchers also thank the ANR for a grant entitled OptChemCom (grant N° ANR-18-CE19-0024-02). This work was supported by the Fondation Leducq in the frame of its program of ERPT equipment support (purchase of an automated patch-clamp system), by a grant "New Team" of the Région Pays de la Loire to M. De Waard, the National Institutes of Health (R01NS091352 to F.B.), by an AFM grant for LED equipment (AFM22401) and by a European FEDER grant in support of the automated patch-clamp system of Nanion. The salary of S. Nicolas was supported by the Fondation Leducq. The fellowships of J. Montnach and L. Filipis are provided by the ANR OptChemCom. The scholarship of L.A. Blömer is provided by the laboratory of excellence "Ion Channels, Science and Therapeutics".

## Author contributions

JM, LL, and SN performed cell culture, automated patch-clamp, RP-HPLC experiments and analyses. FB and DC performed chimera experiments. CC and RB performed the chemical syntheses peptides. CJ performed zebrafish experiments. CH, JM, and AL performed in vivo experiments. HM and CL performed NMR experiments. LAB and LF performed experiments and analyses in brain slices. JM, MC and MDW made the figures. MDW conceived and managed the project. JM, MDW, and MC wrote the manuscript.

## Competing interests

The authors declare no competing interests.
