## [Peer Review File · Nature Communications]

Reviewers' Comments:

Reviewer #1:

Remarks to the Author:

Montnach et al present a novel approach to caging peptide neurotoxins such that they can be activated by light in situ and provide local inhibition of ion channels. Derivatives of the spider toxin Huwentoxin IV were developed with enhanced affinity for Nav1.1, Nav1.2, and Nav1.6 in comparison to its normal target Nav1.7. They attached the photoreactive Nvoc moiety to Lys32, an essential amino acid residue for toxin binding, and showed that inhibition of the target sodium channels was much reduced by Nvoc caging. Analysis of chimeras of Nav1.7 voltage sensor domains implicated Domain II in inhibition of sodium channels, as expected from prior work, and showed that Domain IV chimeras inhibited fast inactivation in the micromolar concentration range. Uncaging released unchanged toxin with full inhibitory activity when tested at 100 nM. Similar Nvoc analogs were produced and tested for hERG and Kv1.2. As proof of principle, Nvoc-HwTxIV was shown to inhibit action potential-dependent sodium influx within 20 to 60 μ m of the center of illumination in brain slices and to impair movement and nerve-muscle function in mice and zebrafish.

General Comments

Nvoc-caged HwTxIV provides proof of the concept that photorelease of the caged polypeptide toxin can inhibit sodium channels locally in neuronal tissue slices and in vivo. This new experimental tool may be valuable in probing electrical excitability in neural circuits in vitro and in vivo. However, although the experiments presented here indicate that the method works in carefully selected experimental settings, Nav1.7-Nvoc has not been used to answer a significant experimental question yet. Because this is a "methods" paper, it would be more appropriate in Nature Methods or a similar journal.

Specific Comments

Lines 106-107. Why target Nav1.1, Nav1.2, and Nav1.6? The native target of HwTxIV, the Nav1.7 channel, is a relevant target for control of chronic pain. The results presented here would be more interesting and practical if the work focused on Nav1.7 and potential uses in pain therapeutics. It is not clear whether inhibition of the other Nav channels studied here would be therapeutically useful.

Figure 4a-c. What is the concentration response curve for uncaged toxin? It appears that the potency of the uncaged HwTxIV may be reduced 10-fold or more compared to wild-type toxin.

Figure 5j, k. The action potential is prolonged in uncaged toxin. Is this caused by block of inactivation? If so, interpretation of the results with this toxin derivative would be difficult because action potentials would be reduced in size, but prolonged at the same time.

Supplementary Figure 3. There are not enough data points to draw fit curves.

Editorial Points

Line 58: peptides

Line 61. Cav2.2

Line 62. "virtually an..."

Line 71. "vision treatment"

Line 81. "targeting voltage gated ion channels more specifically"

Reviewer #2:

Remarks to the Author:

This paper by the De Waard lab presents a proof-of-concept study for an engineered novel photoactivatable spider venom peptide toxin. Nvoc caging effectively blocks the effects of the toxin and photoactivation results in rapid inhibition of sodium currents in vitro and in several model systems. The authors also show that the caging approach works for several other well-known peptide toxins targeting hERG and Kv channels. Their data support that this compound can be activated within a narrow radius of illumination at a selected timepoint after delivery, e.g., spatiotemporal control. Overall, this is a novel approach with broad applicability for development of new research tools in the ion channel field. The potential for therapeutic applications is also there in the longer term, clearly dependent on overcoming multiple challenges, including the development of subtype specific toxins.

Detailed comments

In its present form, the manuscript lacks sufficient detail in the methods for the work to be reproduced, particularly in the structural modeling and synthesis sections, as detailed below.

Throughout the manuscript, the variation in terminology (HwTxIV-Nvoc analogue, HwTxIV-Nvoc, HwTxIV analogue, caged HwTxIV, caged HwTxIV analogue) used for the engineered toxin (HwTxIVG1G4K36-Nvoc) at times makes it difficult to distinguish from the control used for specific experiments.

Line 33Nvoc is not defined/explained anywhere in the manuscript

Line 63-64The statement "Hence, venom peptides appear as the most promising class of compounds..." is not justified as there are obvious and complex issues with peptide toxins as therapeutics. Further, the authors have not compared to any other types of compounds in the manuscript. We suggest revision to "Hence, venom peptides appear as a promising class of compounds..."

Lines 81-82The statement "caged ligands targeting more specifically voltage-gated ion channels remain rare" should probably be rephrased to clarify that only one is known at this time (caged STX).

Line 96It is not clear that the authors actually used the caged peptide toxin to "probe the role of voltage-gated Na⁺ (Nav) channel function." It seems that rather, they used the brain slices to validate the caged toxin.

Line 104It is confusing to have (HwTxIVG1G4K36) here, before the details are presented

Line 109The statement that the HwTxIV-Nav1.7 complex was used appears to contradict the later statement in lines 121-122 and Figure 1 that the modeled complex with NaAb was used for identifying potential modification sites.

Line 112No detail is provided for how the authors arrived at this set of mutations ("design") and no clarification is to be found in either Fig 1 nor Suppl Fig 1. What peptide variants were generated and how were they tested? It would be helpful to have a comparison of the wild-type sequence, perhaps below Fig 1a and to have the mutation sites noted on the structure in Fig 1a.

Line 117Since there were tests of the wild-type and K32N mutant HwTxIV against Nav1.7, why are there no Nav1.7 data for the uncaged analog in Suppl. Fig 1? It would be informative to have this information to compare to the results for the large change toward Nav1.6. Was a similar change observed for Nav1.7?

Lines 112-118There should be a clear indication of what the values are, otherwise the numbers are without meaning. Presumably they are IC₅₀ values for inhibition of sodium currents in the whole-cell patch clamp recordings? It would also be easier to compare the results if they were put into a short table.

Line 121 It should be noted that this interaction is observed in the model that the authors generated as well as the fact that the structure used (6N4R) is a chimera of human Nav1.7 VSD2 with NavAb). In the recent structure of chimeric Nav1.7-NavAb with HwTxIV (PDB ID 7K48), K32 interacts with E815 in the loop between S3 and S4 rather than with E810 in S3. We realize that the 7K48 structure may have been released after the modeling was done, but it should be acknowledged and discussed even if it does not lead to precisely the same prediction for the interference of binding by the Nvoc caging.

Line 211 Did the authors determine which lysine residues were modified in BeKm1-Nvoc and charybdotoxin-Nvoc?

Line 273 On what basis is classical pharmacology “hardly compatible with electrophysiology”?

Line 289 Missing from the discussion are any mention of the drawbacks to using peptide toxins as drugs as well as the challenge of engineering toxins with subtype-specificity which would be essential for development of therapeutics.

Figure 1

Line 688-691 Please clarify the figure panel a and caption text – in some places it says caged HwTxIV-Nvoc and in others it says caged HwTxIV-Nvoc analog. The PDB code and associated publication used for the structure shown in this figure should be indicated as well.

Lines 691-695 Some of these details could be in the methods instead of the caption

Panel c Since the non-caged HwTxIV analog was also purified, can the authors also show the elution profiles for comparison to the caged analog?

Supplementary Figure 1

Panel a It is not clear how informative this docking is because the loop between S3 and S4 was not resolved in this series of structures (6j8g and 6j8h) and the location of the docked HwTxIV shown here does not appear to be consistent with the electron density observed in the structures (see Fig. 2A in <https://pubmed.ncbi.nlm.nih.gov/30765606/>). Note also, this docking is missing from the methods.

Figure 4

Lines 767-771 The PDB models used to generate the molecular figures should be indicated and the references cited

Panel h It would be helpful to have labels that identify which channel is being targeted above the current traces

Methods

Lines 293-299 (Molecular modelling)

- There are insufficient details as to how the model was generated. For example, it should be noted that PDB 6N4R was determined with ProTx2 and the procedure for fitting/modeling the interaction with HwTxIV from 1MB6 should be explained so that others can reproduce the model.
- There is a more recent structure with HwTxIV (PDB 7k48) which may have become available after the initial work, but should still be used as a comparison.
- There is also no information for how the model of HwTxIV on hNav1.7 shown in Supplementary Figure 1 was obtained
- It should be noted that the HwTxIV structure 1MB6 is of the mature toxin and the publication should be cited
- Publications for the other structural models should also be cited

Lines 301-313 (Chemical synthesis)

- This section details the peptide synthesis, but the procedure for addition of the Nvoc group is not included – i.e., is it added during solid phase synthesis? If not, how is the Nvoc group attached to a specific lysine (and not one of the other 4)?
- Details of synthesis of AaHIIR62K-Nvoc, BeKm1-Nvoc, charybdotoxin-Nvoc, including sites of modification for BeKm1, charybdotoxin are missing

Lines 337-350 (Cell cultures and patch clamp)

- The accession codes for the human Nav isoforms are missing
- hNav1.7 is missing from this section and also missing from the patch clamp methods

Lines 389-394 (construction of chimeras)

- There are insufficient details as to how the chimeras were generated. Even though a reference was provided, at least the precise boundaries should be indicated.

Minor suggestions

Line 46 add chronic pain to this list?

Line 55 it would be clearer to replace "toxins" with "animal peptide toxins" (STX and TTX are not peptides)

Line 58 ion channel should be plural

Line 61 should " μ -conotoxin-GVIA" be " ω -conotoxin-GVIA"?

Lines 87, 96 it would be clearer to replace "toxins" with "peptide toxins"

Line 90 "oxytocin, α -conotoxin IMI" should have references (Ref 23 is specific to insulin)

Line 90 "never" should be replaced with "not to our knowledge"

Line 124 would be better to replace "hopefully" with "predicted"

Line 152 "are important enough" would be better as "sufficient"

Line 284 "larvae" should not be plural

Suppl Fig 1, panel b Clarify whether Nav1.1, 1.2, 1.6 were also the human isoforms as suggested by the methods

Reviewer #3:

Remarks to the Author:

This is an outstanding manuscript by Montnach et. al. that describes the development and validation of photoactivatable peptide toxins that target ion channels. The authors focus on a caged variant of HwTxIV, which targets Nav channels, but also extend their approach to the hERG blocker BeKm1 and the Kv1.2 blocker charybdotoxin, as these also contain critical lysine residues that can be readily caged and unmasked using the Nvoc protecting group. The authors extensively characterized HwTxIV-Nvoc using the appropriate chemical and electrophysiological methods and went on to demonstrate its function in brain slices, as well as in vivo. The strategy chosen is very successful and efficient for the following reasons: 1) HwTxIV is very inactive after being caged; 2) uncaging is very clean; 3) caging did not interfere with global peptide folding. Overall the work is of extremely high quality and the methods are well described. The authors' conclusions are well justified by the experiments conducted and the results obtained. Although a fair amount of work has been done with caged peptides in the past, including neuropeptides that activate GPCRs, the development of caged peptide toxins that block ion channels, in many cases, with high specificity is novel and important. The extension to other pharmacologically-specific peptide toxins further increases the impact of this work. The very recent study by Elleman et. al. with caged saxitoxin, is less broad in scope and does not decrease the novelty of this work.

Major concerns

1. Reversibility is not characterized in any experiment. These high affinity ligands likely exhibit slow dissociation rates, but at no point is this addressed. This is important for end-user considerations. The rate of reversal is likely to be context-dependent (cell culture vs brain slice vs in vivo) due to differences in diffusional clearance. Minimally, reversal should be addressed in cell culture experiments in which the uncaged toxin is washed out after photolysis. Better would be to address this in slices, where most applications may occur, and where additional diffusional barriers are present. The data may already exist, to some extent, in the experiment presented in figure 5g, in which action potentials in the same neuron appear to recover after partial block from off-target uncaging. Ideally, we would see the reversal time-course after direct somatic uncaging.

Minor concerns

2. The authors rationalize the impact of the Nvoc caging group in terms of steric impediment to binding. Yet by masking the protonated lysine as an electrostatically neutral carbamate, it also

eliminates a positive charge that is likely critical for binding. Indeed, the K32-to-N mutation leads to complete activity loss, although this variant is sterically similar at position 32. Instead, it simply lacks a positive charge. The charge loss may be of equal importance – it is possible that alkylation with a nitrobenzyl group, for example, may not have reduced affinity as effectively as carbamylation. It is therefore suggested that the authors discuss the role of the positive charge in addition to steric fit.

Responses to reviewer comments

Reviewer #1

General comment: Montnach et al present a novel approach to caging peptide neurotoxins such that they can be activated by light in situ and provide local inhibition of ion channels. Derivatives of the spider toxin Huwentoxin IV were developed with enhanced affinity for Nav1.1, Nav1.2 and Nav1.6 in comparison to its normal target Nav1.7. They attached the photoreactive Nvoc moiety to Lys32, an essential amino acid residue for toxin binding, and showed that inhibition of the target sodium channels was much reduced by Nvoc caging. Analysis of chimeras of Nav1.7 voltage sensor domains implicated Domain II in inhibition of sodium channels, as expected from prior work, and showed that Domain IV chimeras inhibited fast inactivation in the micromolar concentration range. Uncaging released unchanged toxin with full inhibitory activity when tested at 100 nM. Similar Nvoc analogs were produced and tested for hERG and Kv1.2. As proof of principle, Nvoc-HwTxIV was shown to inhibit action potential-dependent sodium influx within 20 to 60 μ m of the center of illumination in brain slices and to impair movement and nerve-muscle function in mice and zebrafish.

Nvoc-caged HwTxIV provides proof of the concept that photorelease of the caged polypeptide toxin can inhibit sodium channels locally in neuronal tissue slices and in vivo. This new experimental tool may be valuable in probing electrical excitability in neural circuits in vitro and in vivo. However, although the experiments presented here indicate that the method works in carefully selected experimental settings, Nav1.7-Nvoc has not been used to answer a significant experimental question yet. Because this is a "methods" paper, it would be more appropriate in Nature Methods or a similar journal.

Response: HwTx-IV was initially discovered for its ability to block, more or less selectively, the Nav1.7 pain target. However, this peptide suffers from important side effects on the neuromuscular junction to become a full antinociceptive agent (mainly by acting on Nav1.6). What we did was to produce an analogue that acts like a pan-Nav channel inhibitor (we deliberately worsened the selectivity properties of this peptide) and now, by modifying the peptide with a photolabile group, the selectivity is provided by the spatio-temporal release of the peptide in a given tissue. Hence, this photoactivable pan-Nav blocker can be used in several types of applications. The experimental question that we addressed here is the following: is it possible to confer selectivity of action to a peptide in vivo that lacks selectivity in vitro? Our peptide acts equally well on Nav1.1, Nav1.2, Nav1.6 and Nav1.7, but we demonstrate that with this technology in hand we gain selectivity for Nav1.6 when we photorelease the peptide in muscles. The same may apply for pain treatment to act on Nav1.7 for intrathecal illumination probably. This technology avoids negative/toxic side effects of peptides lacking selectivity which is an important achievement. This issue of selectivity is now a bit better discussed to clarify this point.

Comment 1: Lines 106-107. Why target Nav1.1, Nav1.2, and Nav1.6? The native target of HwTxIV, the Nav1.7 channel, is a relevant target for control of chronic pain. The results presented here would be more interesting and practical if the work focused on Nav1.7 and potential uses in pain therapeutics. It is not clear whether inhibition of the other Nav channels studied here would be therapeutically useful.

Response: Part of the answer is given above. HwTx-IV by itself is not an ideal compound to target Nav1.7 because of the off-target effects of the peptide. As a result, in spite of several pharmaceutical companies working on improving this peptide, it never made it to the clinics for pain treatment. So why studying pain rather than any other pathology or condition? With this technology, you almost no longer need to have a selective peptide in hand to target a given pathology. Selectivity is provided by the spatio-temporal resolution of the peptide provided that the distributions of the targets are different in the tissue. We added these considerations now in the manuscript because they are important. What we show here is that we can paralyze a muscle locally by light. Botulinum toxins have generated billions in revenues in the cosmetic area by providing local facial paralyses. Herein we provide a hint that a similar form of chosen paralysis can be induced by a much smaller peptide now. But we also demonstrate that the same tool can be used to decipher signaling pathways in neurons, or even alternatively to treat pain as suggested here by the reviewer (this is now stated in the discussion as well). Many other applications can be envisioned indeed. We tried to reinforce our discussion on this point since these advantages were maybe not sufficiently described in the initial version.

Comment 2: Figure 4a-c. What is the concentration response curve for uncaged toxin? It appears that the potency of the uncaged HwTxIV may be reduced 10-fold or more compared to wild-type toxin.

Response: The concentration response-curve is and was given in Figure 3g. The potency of the uncaged toxin is exactly identical to the potency of the non-caged toxin. It is no longer a wild-type HwTx-IV by the way because we worked with an analogue of HwTx-IV. For Figure 4b please bear in mind that illumination was for a given time and given power that not necessarily uncages all the starting compound which may give the impression, based on kinetics of inhibition, that the potency is a bit lower. But this is only an apparent effect. We do not release fully 100 nM and this is not necessary. This is now explained in the legend of Figure 4b.

Comment 3: Figure 5j, k. The action potential is prolonged in uncaged toxin. Is this caused by block of inactivation? If so, interpretation of the results with this toxin derivative would be difficult because action potentials would be reduced in size, but prolonged at the same time.

Response: The shape of the action potential is the result of the combined synergistic activation, de-activation and inactivation of Na⁺ and K⁺ channels. In particular, the re-polarization and therefore the duration of the action potential are regulated by the K⁺ channels activated by the depolarization produced by Na⁺ channel activation. Thus, if depolarization is reduced, the activation of K⁺ channels is also reduced and the action potential is prolonged. This phenomenon is what determines the so-called “burst accommodation”, occurring in many excitatory neurons including pyramidal neurons. As shown in the example below, recorded from the same type of pyramidal neurons used in this study, a high-frequency burst of 3 action potentials is characterized by a progressive decrease of the size of the action potentials associated with a progressive prolongation. The partial block of voltage-gated Na⁺ channels, resulting in a decrease of depolarization, always leads to a physiological prolongation of the action potential.

The physiological phenomenon of action potential prolongation described above is evident in the traces reported in panel d and in panel g of Figure 5. The reviewer, however, mentions panel j. In this case, the action potential is fully blocked, but to test whether also voltage-gated sodium channels were fully blocked we increased the current injected through the patch pipette in order to reach the same depolarization level of the action potential before uncaging. It is important to understand that in this experiment the precise measurement of Na⁺ channel activation was done directly using Na⁺ imaging and not with the indirect measurement of the action potential. Hence, a lower depolarization in this case could simply mean that Na⁺ channel activation is shifted to more positive value, whereas we unambiguously demonstrate that the optically measured Na⁺ influx is fully blocked even when the membrane potential is above +20 mV, which is about the level reached during the action potential. The purple electrical trace in panel j is therefore not an action potential, but the membrane potential resulting from blocking all Na⁺ channels when the cell is depolarized to ~20 mV. We hope that this concept is now clear. We slightly modified the legend of Figure 5 to clarify this issue.

Comment 4: Supplementary Figure 3. There are not enough data points to draw fit curves.

Response: We agree. We removed the fits to the data. We also added some more data points that we had in hand at lower concentrations.

Editorial Points (-> responses)

Line 58: peptides (-> this is what we have in the initial version – we now put “peptide toxins”)

Line 61. Cav2.2 (-> yes modified)

Line 62. "virtually an..." (-> yes modified)

Line 71. "vision treatment" (-> yes agreed – thank you)

Line 81. "targeting voltage gated ion channels more specifically" (-> yes thank you)

Reviewer #2

General comment: This paper by the De Waard lab presents a proof-of-concept study for an engineered novel photoactivatable spider venom peptide toxin. Nvoc caging effectively blocks the effects of the toxin and photoactivation results in rapid inhibition of sodium currents in vitro and in several model systems. The authors also show that the caging approach works for several other well-known peptide toxins targeting hERG and Kv channels. Their data support that this compound can be activated within a narrow radius of illumination at a selected timepoint after delivery, e.g., spatiotemporal control. Overall, this is a novel approach with broad applicability

for development of new research tools in the ion channel field. The potential for therapeutic applications is also there in the longer term, clearly dependent on overcoming multiple challenges, including the development of subtype specific toxins.

Response: We thank the reviewer for the summary of the work. As developed in our response to Reviewer 1, there is also potential for therapeutic applications even if the toxin is not subtype specific. Here, we purposely developed a pan-Nav blocker to demonstrate that selectivity is provided by the spatio-temporal resolution of the uncaging. However, indeed, it remains preferable for therapeutic applications to use channel subtype-specific toxins. There are quite a few published candidate molecules that would fit this requirement. This question of selectivity is now better illustrated in the discussion.

Comment 1: In its present form, the manuscript lacks sufficient detail in the methods for the work to be reproduced, particularly in the structural modeling and synthesis sections, as detailed below.

Throughout the manuscript, the variation in terminology (HwTxIV-Nvoc analogue, HwTxIV-Nvoc, HwTxIV analogue, caged HwTxIV, caged HwTxIV analogue) used for the engineered toxin (HwTxIVG1G4K36-Nvoc) at times makes it difficult to distinguish from the control used for specific experiments.

Response: We agree. We simplified the terminology to ease the reading of the manuscript. The analogue we produced is called “HwTxIV” for simplicity. This peptide is defined in three ways: 1- non-caged HwTxIV (the synthetic peptide without modifications), 2- caged HwTxIV (also called HwTxIV-Nvoc), and 3- uncaged HwTxIV (that represents HwTxIV after removal of Nvoc). This should simplify the reading greatly.

Comment 2: Line 33 - Nvoc is not defined/explained anywhere in the manuscript.

Response: Nvoc is now defined in the text the first time it is used in the introduction.

Comment 3: Line 63-64 - The statement “Hence, venom peptides appear as the most promising class of compounds...” is not justified as there are obvious and complex issues with peptide toxins as therapeutics. Further, the authors have not compared to any other types of compounds in the manuscript. We suggest revision to “Hence, venom peptides appear as a promising class of compounds...”.

Response: We agree. Unnecessary excess of enthusiasm from our side. The text has been modified.

Comment 4: Lines 81-82 - The statement “caged ligands targeting more specifically voltage-gated ion channels remain rare” should probably be rephrased to clarify that only one is known at this time (caged STX).

Response: Yes, indeed. We introduced one sentence in particular to better highlight this important contribution.

Comment 5: Line 96 - It is not clear that the authors actually used the caged peptide toxin to “probe the role of voltage-gated Na⁺ (Nav) channel function.” It seems that rather, they used the brain slices to validate the caged toxin.

Response: Yes, it is probably more accurate to revert the sentence. We rephrased this part.

Comment 6: Line 104 - It is confusing to have (HwTxIVG1G4K36) here, before the details are presented.

Response: Granted. It has been removed from that location and is presented a few lines below.

Comment 7: Line 109 - The statement that the HwTxIV-Nav1.7 complex was used appears to contradict the later statement in lines 121-122 and Figure 1 that the modeled complex with NaAb was used for identifying potential modification sites.

Response: For the search of a new HwTx-IV analogue, we based our work mainly on earlier SAR investigations. We rephrased the initial statement and also reference the work from Wisedchaisri published in *Molecular Cell* in 2021.

Comment 8: Line 112 - No detail is provided for how the authors arrived at this set of mutations (“design”) and no clarification is to be found in either Fig 1 nor Suppl Fig 1. What peptide variants were generated and how were they tested? It would helpful to have a comparison of the wild-type sequence, perhaps below Fig 1a and to have the mutation sites noted on the structure in Fig 1a.

Response: We have introduced a comparison between the wild-type sequence and the analogue in Figure 1a, highlighting the mutations, as requested. This is a good idea indeed. We tested close to 20 variants in a study searching for better Nav1.6 analogues. A manuscript has been submitted to *Frontiers in Cell and Developmental Biology* on this topic. Providing more details on how we selected this mutant would i) denature this publication in *Frontiers*, ii) be out of subject with regard to the technology that would work with any of these analogues, and foremost iii) take unnecessary space in this manuscript. However, we are now more explicit in the result section on what experimental steps were undertaken to define this mutant. If the reviewer needs a copy of this submitted manuscript we can send it upon request.

Comment 9: Line 117 - Since there were tests of the wild-type and K32N mutant HwTxIV against Nav1.7, why are there no Nav1.7 data for the uncaged analog in Suppl. Fig 1? It would be informative to have this information to compare to the results for the large change toward Nav1.6. Was a similar change observed for Nav1.7?

Response: Yes, this is correct. Since we did not target Nav1.7 in any of our experiments, we initially did not show any data on this channel type. However, since most people interested in HwTx-IV are also interested in Nav1.7, we agree that it is informative to add these data as well, opening new opportunities in the field of pain treatment. We have now done these experiments and they have been introduced where most appropriate (Suppl. Figure 1 for non-caged analogue and Figure 2 for caged analogue). This expands the characterization of this analogue to 4 different Nav subtypes, all relevant to neuronal functions. For your information, Supplementary Figure 1 never referred to “uncaged”, but only to HwTxIV. We have also the data for the uncaged HwTxIV that has similar IC50 value on Nav1.7 than HwTxIV itself. See this figure for 3 compounds on hNav1.7: HwTxIV, caged HwTxIV and uncaged HwTxIV:

Comment 10: Lines 112-118 - There should be a clear indication of what the values are, otherwise the numbers are without meaning. Presumably they are IC₅₀ values for inhibition of sodium currents in the whole-cell patch clamp recordings? It would also be easier to compare the results if they were put into a short table.

Response: Yes, sorry. All are IC₅₀ values indeed. This is now indicated in the text. All these data have also been included into a Table as suggested (Supplementary Table 1). Excellent idea indeed.

Comment 11: Line 121 - It should be noted that this interaction is observed in the model that the authors generated as well as the fact that the structure used (6N4R) is a chimera of human Nav1.7 VSD2 with NavAb). In the recent structure of chimeric Nav1.7-NavAb with HwTxIV (PDB ID 7K48), K32 interacts with E815 in the loop between S3 and S4 rather than with E810 in S3. We realize that the 7K48 structure may have been released after the modeling was done, but it should be acknowledged and discussed even if it does not lead to precisely the same prediction for the interference of binding by the Nvoc caging.

Response: Indeed, at the time of modeling, the PDB access code 7K48 was not accessible. We thus performed a modeling work based on the 6N4R structure of the chimeric Nav1.7 VSD2. The modeling that we performed initially is of course less satisfactory than using directly the 7K48 structure for one main reason: the 6N4R structure is lacking the HwTxIV structure obliging to perform a docking exercise. In contrast, the 7K48 structure comprises the structure of HwTxIV. For these reasons, we decided to use for the entire manuscript the data and conclusions generated by the 7K48 structure without including additional docking steps. The numbering of the Nav1.7 channel residues in interaction with K32 has been reproduced from the text of the manuscript from the 10.1016/j.chembiol.2019.10.011 Molecular Cell reference of 2021 and not from the numbering of the PDB file (for some reason there is a shift of -1 between the two). Please note that the E815 is in fact a D815 in PDB and a D816 in the manuscript. We will therefore refer to D816. In the manuscript suggested by the reviewer, K32 interacts with E811 (our E810 in the former version of the manuscript), D816 and E818. For all these reasons, i) we rephrased the interaction of K32 with the Nav1.7 channel residues, ii) panel b of Figure 1 is now based on the 7K48 structure as suggested, and iii) Supplementary Figure 1 panel a is now also based on the 7K48 structure. Of course, with this structure in hand we also identify important steric clashes due to Nvoc addition (also rephrased for accuracy of description).

Comment 12: Line 211 - Did the authors determine which lysine residues were modified in BeKm1-Nvoc and charybdotoxin-Nvoc?

Response: Yes, we did and provide this information now in the text. We also gave more experimental details on the chemical synthesis of these compounds as the Nvoc is already grafted on the chosen lysine residue before we undertake the chemical synthesis of the peptide. It is therefore not a random process after toxin production. We based ourselves on earlier SAR studies mainly.

Comment 13: Line 273 - On what basis is classical pharmacology “hardly compatible with electrophysiology”?

Response: We were referring to the interpretation of electrical signals in brain slices, for instance about the generation or propagation of an action potential, and the fact that toxin application will lead to diffusion in the tissue hampering the interpretation of where the channels are located. However as stated here the sentence is indeed hard to interpret. We simply removed this statement.

Comment 14: Line 289 - Missing from the discussion are any mention of the drawbacks to using peptide toxins as drugs as well as the challenge of engineering toxins with subtype-specificity which would be essential for development of therapeutics.

Response: We added a warning sentence on the issue that peptides can be expensive to produce, and that in some cases their half-lives are short (although again the literature shows very comfortable pharmacokinetics for peptide toxins). Also, they can hardly be delivered orally. We also discussed this matter of subtype-specificity, which may be desirable, but not necessarily, as subtype-specificity can be provided in part by the spatio-temporal characteristics of toxin uncaging (provided that there is not too much local diffusion and that channel subtypes are distributed in different tissues, which is not always the case of course).

Comment 15: Figure 1 Line 688-691 - Please clarify the figure panel a and caption text – in some places it says caged HwTxIV-Nvoc and in others it says caged HwTxIV-Nvoc analog. The PDB code and associated publication used for the structure shown in this figure should be indicated as well.

Lines 691-695 - Some of these details could be in the methods instead of the caption. Panel c: Since the non-caged HwTxIV analog was also purified, can the authors also show the elution profiles for comparison to the caged analog?

Response: Yes, we homogenized the terminology as explained earlier (the analogue is now simply HwTxIV; the original non-modified HwTxIV is termed nHwTxIV for native HwTxIV). The PDB code is provided and the article is cited now. We moved the RP-HPLC experimental conditions to the Methods section as requested. The elution profile of the purified non-caged HwTx-IV have been added in panel c (black trace). The mass data of non-caged HwTx-IV were already present in figure 3 panel d.

Comment 16: Supplementary Figure 1 - Panel a - It is not clear how informative this docking is because the loop between S3 and S4 was not resolved in this series of structures (6j8g and 6j8h) and the location of the docked HwTxIV shown here does not appear to be consistent with the electron density observed in the structures (see Fig. 2A in

<https://pubmed.ncbi.nlm.nih.gov/30765606/>). Note also, this docking is missing from the methods.

Response: There is no longer any docking and as answered to comment 11 we decided therefore to replace it with the proposed 7K48 structure.

Comment 17: Figure 4 - Lines 767-771 - The PDB models used to generate the molecular figures should be indicated and the references cited. Panel h: It would be helpful to have labels that identify which channel is being targeted above the current traces.

Response: The PDB models are now provided for each toxin and appropriate references cited. On panel h the targeted channels are now indicated above current traces.

Comment 18: Methods - Lines 293-299 (Molecular modelling)

- There are insufficient details as to how the model was generated. For example, it should be noted that PDB 6N4R was determined with ProTx2 and the procedure for fitting/modeling the interaction with HwTxIV from 1MB6 should be explained so that others can reproduce the model.
- There is a more recent structure with HwTxIV (PDB 7k48) which may have become available after the initial work, but should still be used as a comparison.
- There is also no information for how the model of HwTxIV on hNav1.7 shown in Supplementary Figure 1 was obtained.
- It should be noted that the HwTxIV structure 1MB6 is of the mature toxin and the publication should be cited.
- Publications for the other structural models should also be cited.

Response: We have now changed the Methods section referring to the use of the 7K48 structure. We no longer do modeling. Supplementary Figure 1 text has been modified accordingly and we cited the publications where appropriate.

Comment 19: Lines 301-313 (Chemical synthesis).

- This section details the peptide synthesis, but the procedure for addition of the Nvoc group is not included – i.e., is it added during solid phase synthesis? If not, how is the Nvoc group attached to a specific lysine (and not one of the other 4)?
- Details of synthesis of AaHIIR62K-Nvoc, BeKm1-Nvoc, charybdotoxin-Nvoc, including sites of modification for BeKm1, charybdotoxin are missing.

Response: Yes, we should have added this information earlier. The Nvoc group is already attached to the fmoc-Lys residue before the chemical synthesis starts. This is now stated and the origin of the modified Lys residue provided. By proceeding this way, we choose the lysine residue to be modified in the sequence. The chemical syntheses of AaHIIR62K-Nvoc, BeKm1-Nvoc and charybdotoxin-Nvoc are now briefly described and respect the same rules than the synthesis of HwTx-IV-Nvoc.

Comment 20: Lines 337-350 (Cell cultures and patch clamp).

- The accession codes for the human Nav isoforms are missing.
- hNav1.7 is missing from this section and also missing from the patch clamp methods.

Response: Yes, indeed. The hNav1.7 cell line and patch clamp methods are now described as well. The accession codes provided.

Comment 21: Lines 389-394 (construction of chimeras)
• There are insufficient details as to how the chimeras were generated. Even though a reference was provided, at least the precise boundaries should be indicated.

Response: The details about the precise boundaries are now provided by including the exact amino acid sequences that were swapped.

Comment 22: Minor suggestions.

Line 46 : add chronic pain to this list?

Line 55: it would be clearer to replace “toxins” with “animal peptide toxins” (STX and TTX are not peptides).

Line 58: ion channel should be plural.

Line 61: should “ μ -conotoxin-GVIA” be “ ω -conotoxin-GVIA”?

Lines 87, 96: it would be clearer to replace “toxins” with “peptide toxins”.

Line 90: “oxytocin, α -conotoxin IMI” should have references (Ref 23 is specific to insulin).

Line 90: “never” should be replaced with “not to our knowledge”.

Line 124: would be better to replace “hopefully” with “predicted”.

Line 152: “are important enough” would be better as “sufficient”.

Line 284: “larvae” should not be plural.

Suppl Fig 1, panel b: Clarify whether Nav1.1, 1.2, 1.6 were also the human isoforms as suggested by the methods.

Response: We performed all the suggested modifications. Yes in Suppl. Fig. 1, these were all human clones. Reference 23 is valid for all the three peptides and is now referenced several times.

Reviewer #3

General comment: This is an outstanding manuscript by Montnach et. al. that describes the development and validation of photoactivatable peptide toxins that target ion channels. The authors focus on a caged variant of HwTxIV, which targets Nav channels, but also extend their approach to the hERG blocker BeKm1 and the Kv1.2 blocker charybdotoxin, as these also contain critical lysine residues that can be readily caged and unmasked using the Nvoc protecting group. The authors extensively characterized HwTxIV-Nvoc using the appropriate chemical and electrophysiological methods and went on to demonstrate its function in brain slices, as well as in vivo. The strategy chosen is very successful and efficient for the following reasons: 1) HwTxIV is very inactive after being caged; 2) uncaging is very clean; 3) caging did not interfere with global peptide folding. Overall the work is of extremely high quality and the methods are well described. The authors' conclusions are well justified by the experiments conducted and the results obtained. Although a fair amount of work has been done with caged peptides in the past, including neuropeptides that activate GPCRs, the development of caged peptide toxins that block ion channels, in many cases, with high specificity is novel and important. The extension to other pharmacologically-specific peptide toxins further increases the impact of this work. The very recent study by Elleman et. al. with caged saxitoxin, is less broad in scope and does not decrease the novelty of this work.

Response: We thank the reviewer for this general appreciation of the work.

Comment 1: Major concerns:

1. Reversibility is not characterized in any experiment. These high affinity ligands likely exhibit slow dissociation rates, but at no point is this addressed. This is important for end-user considerations. The rate of reversal is likely to be context-dependent (cell culture vs brain slice vs *in vivo*) due to differences in diffusional clearance. Minimally, reversal should be addressed in cell culture experiments in which the uncaged toxin is washed out after photolysis. Better would be to address this in slices, where most applications may occur, and where additional diffusional barriers are present. The data may already exist, to some extent, in the experiment presented in figure 5g, in which action potentials in the same neuron appear to recover after partial block from off-target uncaging. Ideally, we would see the reversal time-course after direct somatic uncaging.

Response: Peptide toxins differ quite drastically in their rate of reversal. Some are easily washed out (BeKm-1 or charybdotoxin), while, according to its properties (lipid affinity (Agwa et al. *J. Biol. Chem.* 295, 5067-5080, 2020)), HwTx-IV is expected to have a slow rate of reversal. We agree with the reviewer that the rate of reversal should be context-dependent: faster in cell cultures and slower in brain slices and *in vivo* because of diffusional barriers. We have now performed experiments that address this issue for caged HwTx-IV both in patch clamp experiments and in tissue slices. We focused mainly on hNav1.6 since HwTx-IV binding site on hNav1.1 and hNav1.2 is well-preserved. We thus do not expect to see differences in behavior on these channel subtypes compared to hNav1.6. As expected from earlier studies, we do confirm that, after washing the extracellular buffer, we recover only a small percentage of current after photolysis of caged HwTxIV. The data have been added in the supplementary data and have been discussed (one supplementary figure on patch clamp data and another in brain slices). The issue of reversibility is essential and depends on the application pursued. For researchers that aim to look at recovery, or repeatability of the light exposure, or simply wish a transient pharmacological response, that can be repeated with multiple light exposures, then obviously reversibility is an asset. In the application we developed (selective muscle immobilization), then lack of reversibility is the preferred option. This issue of reversibility was more or less addressed in Figure 6e where 10 min after the arrest of illumination, the muscle is still inhibited in contraction (illustrating little reversibility *in vivo* for this particular toxin). In the case of BeKm-1, that is reversible, and that acts on hERG channels, it may be of interest to use this probe to induce reversible arrhythmias *in vivo* in a repeated fashion. We are working on this application for a future publication.

Here is a figure taken from our recent publication on the use of BeKm-1 on cardiomyocytes derived from human iPS showing the reversible action of BeKm-1. So, reversibility is a matter of choice (by choosing the appropriate toxin) and a question of application. This is now also discussed in the article since we agree with the reviewer that it is an essential question.

Comment 2: Minor concerns.

2. The authors rationalize the impact of the Nvoc caging group in terms of steric impediment to binding. Yet by masking the protonated lysine as an electrostatically neutral carbamate, it also eliminates a positive charge that is likely critical for binding. Indeed, the K32-to-N mutation leads to complete activity loss, although this variant is sterically similar at position 32. Instead, it simply lacks a positive charge. The charge loss may be of equal importance – it is possible that alkylation with a nitrobenzyl group, for example, may not have reduced affinity as effectively as carbamylation. It is therefore suggested that the authors discuss the role of the positive charge in addition to steric fit.

Response: we definitively agree with the reviewer and this possibility is now discussed as well.

Reviewers' Comments:

Reviewer #1:

Remarks to the Author:

The authors have responded appropriately to my concerns. I recommend acceptance for publication.

Reviewer #2:

Remarks to the Author:

I thank the authors for the superb job they have done at addressing all of my concerns. Congratulations on a wonderful study!!

Rajesh Khanna, PhD
University of Arizona

Reviewer #3:

Remarks to the Author:

My concerns have been fully addressed. This manuscript is ready for publication.